# Gating is Weighting: Understanding Gated Linear Attention through In-context Learning

**Yingcong Li**[*]
University of Michigan
yingcong@umich.edu

**Davoud Ataee Tarzanagh**[*]
Samsung SDS Research America
d.tarzanagh@samsung.com

**Ankit Singh Rawat**
Google Research NYC
ankitsrawat@google.com

**Maryam Fazel**
University of Washington
mfazel@uw.edu

**Samet Oymak**
University of Michigan
oymak@umich.edu

## Abstract

Linear attention methods offer a compelling alternative to softmax attention due to their efficiency in recurrent decoding. Recent research has focused on enhancing standard linear attention by incorporating gating while retaining its computational benefits. Such Gated Linear Attention (GLA) architectures include highly competitive models such as Mamba and RWKV. In this work, we investigate the in-context learning capabilities of the GLA model and make the following contributions. We show that a multilayer GLA can implement a general class of Weighted Preconditioned Gradient Descent (WPGD) algorithms with data-dependent weights. These weights are induced by the gating mechanism and the input, enabling the model to control the contribution of individual tokens to prediction. To further understand the mechanics of this weighting, we introduce a novel data model with multitask prompts and characterize the optimization landscape of learning a WPGD algorithm. We identify mild conditions under which there exists a unique global minimum, up to scaling invariance, and the associated WPGD algorithm is unique as well. Finally, we translate these findings to explore the optimization landscape of GLA and shed light on how gating facilitates context-aware learning and when it is provably better than vanilla linear attention.

## 1 Introduction

The Transformer (Vaswani, 2017) has become the de facto standard for language modeling tasks. The key component of the Transformer is the self-attention mechanism, which computes softmax-based similarities between all token pairs. Despite its success, the self-attention mechanism has quadratic complexity with respect to sequence length, making it computationally expensive for long sequences. To address this issue, a growing body of work has proposed near-linear time approaches to sequence modeling. The initial approaches included linear attention and state-space models, both achieving $O(1)$ inference complexity per generated token, thanks to their recurrent form. While these initial architectures typically do not match softmax attention in performance, recent recurrent models such as Mamba (Gu & Dao, 2023; Dao & Gu, 2024), mLSTM (Beck et al., 2024), DeltaNet (Yang et al., 2024b;a), GLA Transformer (Yang et al., 2023), and RWKV-6 (Peng et al., 2024) achieve highly competitive results with the softmax Transformer. Notably, as highlighted in Yang et al. (2023), these architectures can be viewed as variants of *gated linear attention* (GLA), which incorporates a gating mechanism within the recurrence of linear attention. Additionally, Behrouz et al. (2025a;b) unify those models as associative memory modules that optimize internal objectives through iterative algorithms, revealing connections to online optimization and memory management dynamics.

---

[*]Equal contribution.

Given a sequence of tokens $(z_i)_{i=1}^{n+1} \in \mathbb{R}^{d+1}$ and the associated query, key, and value embeddings $(q_i, k_i, v_i)_{i=1}^{n+1} \subset \mathbb{R}^{d+1}$, with $d+1$ being the embedding dimension[1], the GLA recurrence is given by

$$S_i = G_i \odot S_{i-1} + v_i k_i^\top, \quad \text{and} \quad o_i = S_i q_i, \quad i \in \{1, \dots, n+1\}. \tag{GLA}$$

Here, $S_i \in \mathbb{R}^{(d+1) \times (d+1)}$ represents the *2D state variable* with $S_0 = 0$; $o_i \in \mathbb{R}^{d+1}$ represents the $i$th output token; and the *gating variable* $G_i := G(z_i) \in \mathbb{R}^{(d+1) \times (d+1)}$ is applied to the state $S_{i-1}$ through the Hadamard product $\odot$ and $G$ represents the gating function. When $G_i$ is a matrix of all ones, (GLA) reduces to causal linear attention (Katharopoulos et al., 2020).

The central objective of this work is to enhance the mathematical understanding of the GLA mechanism. In-context learning (ICL), one of the most remarkable features of modern sequence models, provides a powerful framework to achieve this aim. ICL refers to the ability of a sequence model to implicitly infer functional relationships from the demonstrations provided in its context window (Brown, 2020; Min et al., 2022). It is inherently related to the model's ability to emulate learning algorithms. Notably, ICL has been a major topic of empirical and theoretical interest in recent years. For example, Ali et al. (2024) show that Mamba embeds implicit attention matrices via data-controlled linear operators, while Sieber et al. (2024) unify SSMs, attention, and RNNs within a dynamical-systems framework. More specifically, a series of works have examined the approximation and optimization characteristics of linear attention, and have provably connected linear attention to the preconditioned gradient descent algorithm (Von Oswald et al., 2023; Ahn et al., 2024; Zhang et al., 2024). Given that the GLA recurrence in (GLA) has a richer design space, this leads us to ask:

*What learning algorithm does GLA emulate in ICL?*

**Contributions:** The (GLA) recurrence enables the sequence model to weight past information in a data-dependent manner through the gating mechanism $(G_i)_{i=1}^n$. Building on this observation, we demonstrate that a GLA model can implement a *data-dependent Weighted Preconditioned Gradient Descent (WPGD)* algorithm. Specifically, a one-step WPGD with scalar gating, where all entries of $G_i$ are identical, is described by the prediction:

$$\hat{y} = x^\top P X^\top (y \odot \omega). \tag{1}$$

Here, $X \in \mathbb{R}^{n \times d}$ is the input feature matrix; $y \in \mathbb{R}^n$ is the associated label vector; $x \in \mathbb{R}^d$ represents the test/query input to predict; $P \in \mathbb{R}^{d \times d}$ is the preconditioning matrix; and $\omega \in \mathbb{R}^n$ weights the individual samples. When $\omega$ is fixed, we drop "data-dependent" and simply refer to this algorithm as the WPGD algorithm. However, for GLA, $\omega$ depends on the data through recursive multiplication of the gating variables. Building on this formalism, we make the following specific contributions:

◇ **ICL capabilities of GLA (§3):** Through constructive arguments, we demonstrate that a multilayer GLA model can implement data-dependent WPGD iterations, with weights induced by the gating function. This construction sheds light on the role of causal masking and the expressivity distinctions between scalar- and vector-valued gating functions.

◇ **Landscape of one-step WPGD (§4):** The GLA⇔WPGD connection motivates us to ask: *How does WPGD weigh demonstrations in terms of their relevance to the query?* To address this, we study the fundamental problem of learning an optimal WPGD algorithm: Given a tuple $(X, y, x, y) \sim \mathcal{D}$, with $y \in \mathbb{R}$ being the label associated with the query, we investigate the population risk minimization:

$$\mathcal{L}_{\mathsf{WPGD}}^\star := \min_{P \in \mathbb{R}^{d \times d}, \omega \in \mathbb{R}^n} \mathcal{L}_{\mathsf{WPGD}}(P, \omega),$$

$$\text{where} \quad \mathcal{L}_{\mathsf{WPGD}}(P, \omega) = \mathbb{E}_\mathcal{D}\left[\left(y - x^\top P X^\top (y \odot \omega)\right)^2\right]. \tag{2}$$

---

[1]The sequence length and embedding dimension are set to $n+1$ and $d+1$ per the prompt definition in (3).

As our primary mathematical contribution, we characterize the loss landscape under a general multitask data setting, where the tasks associated with the demonstrations $(\boldsymbol{X}, \boldsymbol{y})$ have varying degrees of correlation to the target task $(\boldsymbol{x}, y)$. We carefully analyze this loss landscape and show that, under mild conditions, there is a unique global minimum $(\boldsymbol{P}, \boldsymbol{\omega})$ up to scaling invariance, and the associated WPGD algorithm is also unique.

◇ **Loss landscape of one-layer GLA (§5):** The landscape is highly intricate due to the recursively multiplied gating variables. We show that learning the optimal GLA layer can be connected to solving (2) with a constraint $\boldsymbol{\omega} \in \mathcal{W}$, where the restriction $\mathcal{W}$ is induced by the choice of gating function and input space. Solidifying this connection, we introduce a multitask prompt model under which we characterize the loss landscape of GLA and the influence of task correlations. Our analysis and experiments reveal insightful distinctions between linear attention, GLA with scalar gating, and GLA with vector-valued gating.

## 2 Problem setup

*Notations.* $\mathbb{R}^d$ is the $d$-dimensional real space, with $\mathbb{R}_+^d$ and $\mathbb{R}_{++}^d$ as its positive and strictly positive orthants. The set $[n]$ denotes $\{1, \dots, n\}$. Bold letters, e.g., $\boldsymbol{a}$ and $\boldsymbol{A}$, represent vectors and matrices. The identity matrix of size $n$ is denoted by $\boldsymbol{I}_n$. The symbols $\boldsymbol{1}$ and $\boldsymbol{0}$ denote the all-one and all-zero vectors or matrices of proper size, such as $\boldsymbol{1}_d \in \mathbb{R}^d$ and $\boldsymbol{1}_{d \times d} \in \mathbb{R}^{d \times d}$. The subscript is omitted when the dimension is clear from the context. The Gaussian distribution with mean $\boldsymbol{\mu}$ and covariance $\boldsymbol{\Sigma}$ is written as $\mathcal{N}(\boldsymbol{\mu}, \boldsymbol{\Sigma})$. The Hadamard product (element-wise multiplication) is denoted by $\odot$, and Hadamard division (element-wise division) is denoted by $\oslash$. Given any $\boldsymbol{a}_{i+1}, \dots, \boldsymbol{a}_j \in \mathbb{R}^d$, we define $\boldsymbol{a}_{i:j} = \boldsymbol{a}_{i+1} \odot \cdots \odot \boldsymbol{a}_j$ for $i < j$, and $\boldsymbol{a}_{i:i} = \boldsymbol{1}_d$.

The objective of this work is to develop a theoretical understanding of GLA through ICL. The optimization landscape of standard linear attention has been a topic of significant interest in the ICL literature (Ahn et al., 2024; Li et al., 2024b). Following these works, we consider the input prompt

$$\boldsymbol{Z} = \begin{bmatrix} \boldsymbol{z}_1 & \cdots & \boldsymbol{z}_n & \boldsymbol{z}_{n+1} \end{bmatrix}^\top = \begin{bmatrix} \boldsymbol{x}_1 & \cdots & \boldsymbol{x}_n & \boldsymbol{x}_{n+1} \\ y_1 & \cdots & y_n & 0 \end{bmatrix}^\top \in \mathbb{R}^{(n+1) \times (d+1)}, \tag{3}$$

where tokens encode the input-label pairs $(\boldsymbol{x}_i, y_i)_{i=1}^n \subset \mathbb{R}^d \times \mathbb{R}$.

We aim to enable ICL by training a sequence model $F : \mathbb{R}^{(n+1) \times (d+1)} \to \mathbb{R}$ that predicts the label $y := y_{n+1}$ associated with the query $\boldsymbol{x} := \boldsymbol{x}_{n+1}$. This model will utilize the demonstrations $(\boldsymbol{x}_i, y_i)_{i=1}^n$ to infer the mapping between $\boldsymbol{x}$ and $y$. Assuming that the data is distributed as $(y, \boldsymbol{Z}) \sim \mathcal{D}$, the ICL objective is defined as

$$\mathcal{L}(F) = \mathbb{E}_{\mathcal{D}} \left[ (y - F(\boldsymbol{Z}))^2 \right]. \tag{4}$$

**Linear attention and shared-task distribution.** Central to our paper is the choice of the function class $F$. When $F$ is a linear attention model, the prediction $\hat{y}_F := F(\boldsymbol{Z})$ takes the form $\hat{y}_F = \boldsymbol{z}_{n+1}^\top \boldsymbol{W}_q \boldsymbol{W}_k^\top \boldsymbol{Z}^\top \boldsymbol{Z} \boldsymbol{W}_v \boldsymbol{h}$, where $\boldsymbol{W}_k, \boldsymbol{W}_q, \boldsymbol{W}_v \in \mathbb{R}^{(d+1) \times (d+1)}$ are attention parameters, and $\boldsymbol{h} \in \mathbb{R}^{d+1}$ is the linear prediction head.

We assume that the in-context input-label pairs follow a *shared-task distribution*, where $\boldsymbol{\beta} \sim \mathcal{N}(\boldsymbol{0}, \boldsymbol{\Sigma}_\beta)$, $\boldsymbol{x}_i$ are i.i.d. with $\boldsymbol{x}_i \sim \mathcal{N}(\boldsymbol{0}, \boldsymbol{\Sigma}_x)$, and $y_i \sim \mathcal{N}(\boldsymbol{\beta}^\top \boldsymbol{x}_i, \sigma^2)$, where $\sigma \geq 0$ represents the noise level. Under this shared-task distribution, it is shown (Von Oswald et al., 2023; Ahn et al., 2024; Zhang et al., 2024) that the optimal one-layer linear attention prediction coincides with the one-step optimal preconditioned gradient descent (PGD). In particular, considering the data distribution discussed above, we have

$$\hat{y}_F = \boldsymbol{x}^\top \hat{\boldsymbol{\beta}}, \quad \text{where} \quad \hat{\boldsymbol{\beta}} = \boldsymbol{P}^\star \boldsymbol{X}^\top \boldsymbol{y}, \tag{5}$$

and

$$P^\star := \underset{P \in \mathbb{R}^{d \times d}}{\operatorname{argmin}} \; \mathbb{E}\left[\left(y - x^\top P X^\top y\right)^2\right] \quad \text{with} \quad X := [x_1 \; \cdots \; x_n]^\top \quad \text{and} \quad y := [y_1 \; \cdots \; y_n]^\top. \quad (6)$$

**Linear attention and gating.** Given the input prompt $Z$ as in (3), let $(q_i, k_i, v_i) = (W_q^\top z_i, W_k^\top z_i, W_v^\top z_i)$ be the corresponding query, key, and value embeddings. The output of *causal* linear attention at time $i$ can be computed using (GLA) with $G_i = 1$. This recurrent form implies that linear attention has $O(d^2)$ cost, that is independent of sequence length $n$, to generate per-token. GLA follows the same structure as linear attention but with a gating mechanism (i.e., $G_i \neq 1$), which equips the model with the option to pass or supress the history. As discussed in Yang et al. (2023), the different choices of the gating function correspond to different popular recurrent architectures such as Mamba (Gu & Dao, 2023), Mamba2 (Dao & Gu, 2024), RWKV (Peng et al., 2024), etc.

We will show that GLA can weigh the context window through gating, thus, its capabilities are linked to the WPGD algorithm described in (8). This will in turn facilitate GLA to effectively learn *multitask prompt distributions* described by $y_i \sim \mathcal{N}(\beta_i^\top x_i, \sigma^2)$ with $\beta_i$'s not necessarily identical.

## 3 What gradient methods can GLA emulate?

In this section, we investigate the ICL capabilities of GLA and show that under suitable instantiations of model weights, GLA can implement *data-dependent* WPGD.

### 3.1 GLA as a data-dependent WPGD predictor

**Data-Dependent WPGD.** Given $X$ and $y$ as defined in (6), consider the weighted least squares objective $\mathcal{L}(\beta) = \sum_{i=1}^n \omega_i \cdot (y_i - \beta^\top x_i)^2$ with weights $\omega = [\omega_1, \omega_2, \cdots, \omega_n]^\top \in \mathbb{R}^n$. To optimize this, we use gradient descent (GD) starting from zero initialization, $\beta_0 = 0$ with a step size of $\eta = 1/2$. One step of standard GD is given by

$$\beta_1 = \beta_0 - \eta \nabla \mathcal{L}(\beta_0) = \sum_{i=1}^n \omega_i \cdot x_i y_i = X^\top (\omega \odot y).$$

Given a test/query feature $x$, the corresponding prediction is $\hat{y} = x^\top \hat{\beta}$ where $\hat{\beta} = \beta_1$. Additionally, if we were using PGD with a preconditioning matrix $P \in \mathbb{R}^{d \times d}$, $\beta_0 = 0$, and $\eta = 1/2$, then a single iteration results in

$$\hat{y} = x^\top \hat{\beta}, \quad \text{where} \quad \hat{\beta} = \beta_0 - \eta P \nabla \mathcal{L}(\beta_0) = P X^\top (\omega \odot y). \quad (7)$$

Above is the basic *sample-wise* WPGD predictor which weights individual datapoints, in contrast to the PGD predictor as presented in (5). It turns out, *vector-valued gating* can facilitate a more general estimator which weights individual coordinates. To this aim, we introduce an extension as follows: Let $P_1, P_2 \in \mathbb{R}^{d \times d}$ denote the preconditioning matrices, and let $\Omega \in \mathbb{R}^{n \times d}$ denote the *vector-valued weighting* matrix. Note that $\Omega$ is a weight matrix that enables coordinate-wise weighting. Throughout the paper, we use $\omega$ and $\Omega$ to denote sample-wise and coordinate-wise weighting parameters, corresponding to scalar and vector gating strategies, respectively. Then given $\Omega$, we can similarly define

$$\beta_1^{\mathrm{gd}}(P_1, P_2, \Omega) := P_2 (X P_1 \odot \Omega)^\top y \quad (8a)$$

as one-step of (generalized) WPGD. Its corresponding prediction on a test query $x$ is:

$$\hat{y} = x^\top \hat{\beta}, \quad \text{where} \quad \hat{\beta} = \beta_1^{\mathrm{gd}}(P_1, P_2, \Omega). \quad (8b)$$

We note that by removing the weighting matrix $\Omega$, (8) reduces to standard PGD (cf. (5)) and setting $\Omega = \omega 1_d^\top$ reduces to sample-wise WPGD (cf. (7)). Li et al. (2024b) has demonstrated

that H3-like models implement one-step *sample-wise* WPGD, and they focus on the shared-task distribution where $\boldsymbol{\beta}_i \equiv \boldsymbol{\beta}$ for all $i \in [n]$. In contrast, our work considers a more general data setting where tasks within an in-context prompt are not necessarily identical.

We introduce the following model constructions under which we establish the equivalence between GLA (cf. (GLA)) and WPGD (cf. (8)), where the weighting matrix is induced by the input data and the gating function. Inspired by prior works (Von Oswald et al., 2023; Ahn et al., 2024), we consider the following restricted attention matrices:

$$W_k = \begin{bmatrix} P_k & 0 \\ 0 & 0 \end{bmatrix}, \quad W_q = \begin{bmatrix} P_q & 0 \\ 0 & 0 \end{bmatrix}, \quad \text{and} \quad W_v = \begin{bmatrix} 0_{d \times d} & 0 \\ 0 & 1 \end{bmatrix}, \tag{9}$$

where $P_k, P_q \in \mathbb{R}^{d \times d}$. Note that the $(d+1, d+1)$-th entry of $W_v$ is set to one for simplicity. More generally, this entry can take any nonzero value, e.g., $v \in \mathbb{R}$. Parameterizing $W_q$ with $P_q/v$ produces the same output as in (9).

**Theorem 1.** *Recall (GLA) with input sequence $\boldsymbol{Z} = \begin{bmatrix} z_1 & \cdots & z_n & z_{n+1} \end{bmatrix}^\top$ defined in (3), the gating $G(z_i) = \boldsymbol{G}_i \in \mathbb{R}^{(d+1) \times (d+1)}$, and outputs $(\boldsymbol{o}_i)_{i=1}^{n+1}$. Consider model construction in (9) and take the last coordinate of the last token output denoted by $(\boldsymbol{o}_{n+1})_{d+1}$ as a prediction. Then, we have*

$$f_{\mathsf{GLA}}(\boldsymbol{Z}) := (\boldsymbol{o}_{n+1})_{d+1} = \boldsymbol{x}^\top \hat{\boldsymbol{\beta}}, \quad \text{where} \quad \hat{\boldsymbol{\beta}} = \boldsymbol{\beta}_1^{gd}(\boldsymbol{P}_k, \boldsymbol{P}_q, \boldsymbol{\Omega}).$$

*Here, $\boldsymbol{\beta}_1^{gd}(\cdot)$ is a one-step WPGD feature predictor defined in (8a); $\boldsymbol{P}_k$ and $\boldsymbol{P}_q$ correspond to attention weights following (9); and $\boldsymbol{\Omega} = \begin{bmatrix} \boldsymbol{g}_{1:n+1} & \cdots & \boldsymbol{g}_{n:n+1} \end{bmatrix}^\top \in \mathbb{R}^{n \times d}$, where $\boldsymbol{g}_{i:n+1}, i \in [n]$ is given by*

$$\boldsymbol{g}_{i:n+1} := \boldsymbol{g}_{i+1} \odot \boldsymbol{g}_{i+2} \cdots \boldsymbol{g}_{n+1} \in \mathbb{R}^d, \quad \text{and} \quad \boldsymbol{G}_i = \begin{bmatrix} * & * \\ \boldsymbol{g}_i^\top & * \end{bmatrix}. \tag{10}$$

Here and throughout, we use $*$ to fill the entries of the matrices that do not affect the final output. Based on the model construction in (9), these entries can be assigned any value.

Observe that, crucially, since $\boldsymbol{g}_i$ is associated with $z_i$, $z_i$ influences the weighting of $z_j$ for all $j < i$. We defer the proof of Theorem 1 to the Appendix D.1. It is noticeable that only $d$ of the total $(d+1)^2$ entries in each gating matrix $\boldsymbol{G}_i$ are useful due to the model construction presented in (9). However, if we relax the weight restriction, e.g., $\boldsymbol{W}_v = \begin{bmatrix} 0_{(d+1) \times d} & \boldsymbol{u} \end{bmatrix}^\top$, where $\boldsymbol{u} \in \mathbb{R}^{d+1}$, then the weighting matrix $\boldsymbol{\Omega}$ in Theorem 1 is associated with all rows of the $\boldsymbol{G}_i$ matrices. We defer to Eqn. (27) and the discussion in Appendix D.1.

**Capabilities of multi-layer GLA.** Ahn et al. (2024) demonstrated that, with appropriate construction, an $L$-layer linear attention model performs $L$ steps of PGD on the dataset $(\boldsymbol{x}_i, y_i)_{i=1}^n$ provided within the prompt. In Appendix A, we extend this analysis to multi-layer GLA and characterize the algorithmic class it can emulate. Importantly, Ahn et al. (2024) does not account for *causal masking*, which is a fundamental aspect of multi-layer GLA due to its recurrent structure as described in (GLA). Our main result, Theorem 6 in Appendix A, establishes that an $L$-layer GLA implements $L$ steps of WPGD, where the gradients are computed in a recurrent form.

**GLA with scalar gating.** Theorem 1 establishes a connection between one-layer GLA (cf. (GLA)) and one-step WPGD (cf. (8)), where the weighting in WPGD corresponds to the gating $G(z_i) = \boldsymbol{G}_i$ in GLA, as detailed in Theorem 1. Now let us consider the widely used types of gating functions, such as $\boldsymbol{G}_i = \alpha_i \boldsymbol{1}_{d+1}^\top$ (Yang et al., 2023; Katsch, 2023; Qin et al., 2024; Peng et al., 2024) or $\boldsymbol{G}_i = \gamma_i \boldsymbol{1}_{(d+1),(d+1)}$ (Dao & Gu, 2024; Beck et al., 2024; Peng et al., 2021; Sun et al., 2024), where $\alpha_i \in \mathbb{R}^{d+1}$ and $\gamma_i \in \mathbb{R}$. In both cases, the gating matrices in (10) take the form of $\begin{bmatrix} * & * \\ g_i \boldsymbol{1}_d^\top & * \end{bmatrix}$ for some $g_i \in \mathbb{R}$, thus simplifying the predictor to a sample-wise WPGD, as given by

$$f_{\mathsf{GLA}}(\boldsymbol{Z}) = \hat{\boldsymbol{\beta}}^\top \boldsymbol{x}, \quad \text{with} \quad \hat{\boldsymbol{\beta}} = \boldsymbol{P} \boldsymbol{X}^\top (\boldsymbol{\omega} \odot \boldsymbol{y}), \tag{11}$$

where $\boldsymbol{P} = \boldsymbol{P}_q \boldsymbol{P}_k^\top$ and $\boldsymbol{\omega} = \begin{bmatrix} g_{1:n+1} & \cdots & g_{n:n+1} \end{bmatrix}^\top \in \mathbb{R}^n$. In the remainder, we will mostly focus on the one-layer GLA with scalar gating as presented in (11).

## 4 Optimization landscape of WPGD

In this section, we explore the problem of learning the optimal sample-wise WPGD algorithm described in (11), a key step leading to our analysis of GLA. The problem is as follows. Recap from (6) that we are given the tuple $(x, y, X, y) \sim \mathcal{D}$, where $X \in \mathbb{R}^{n \times d}$ is the input matrix, $y \in \mathbb{R}^n$ is the label vector, $x \in \mathbb{R}^d$ is the query, and $y \in \mathbb{R}$ is its associated label. The goal is to use $X, y$ to predict $y$ given $x$ via the one-step WPGD prediction $\hat{y} = x^\top \hat{\beta}$, with $\hat{\beta}$ as in (11). The algorithm learning problem is given by (2) which minimizes the WPGD risk $\mathbb{E}_{\mathcal{D}}[(y - x^\top PX(\omega \odot y))^2]$.

Prior research (Mahankali et al., 2023; Li et al., 2024b; Ahn et al., 2024) has studied the problem of learning PGD when input-label pairs follow an i.i.d. distribution. It is worth noting that while Li et al. (2024b) establishes a connection between H3-like models and (11) similar to ours, their work assumes that the optimal $\omega$ consists of all ones and does not specifically explore the optimization landscape of $\omega$ when in-context samples are not generated from the same task vector $\beta$. Departing from this, we introduce a realistic model where each input-label pair is allowed to come from a distinct task.

**Definition 1** (Correlated Task Model). *Suppose $\beta_i \in \mathbb{R}^d \sim \mathcal{N}(0, I)$ are jointly Gaussian for all $i \in [n+1]$. Define*

$$\beta := \beta_{n+1}, \quad B := [\beta_1 \ \dots \ \beta_n]^\top, \quad R := \frac{1}{d}\mathbb{E}[BB^\top], \quad \text{and} \quad r := \frac{1}{d}\mathbb{E}[B\beta]. \tag{12}$$

Note that in (12), we have $R \in \mathbb{R}^{n \times n}$ and $r \in \mathbb{R}^n$, with normalization ensuring that the entries of $R$ and $r$ lie in the range $[-1, 1]$, corresponding to correlation coefficients. Further, due to the joint Gaussian nature of the tasks, for any $i, j \in [n+1]$, the residual $\beta_i - r_{ij}\beta_j$ is independent of $\beta_j$.

**Definition 2** (Multitask Distribution). *Let $(\beta_i)_{i=1}^{n+1}$ be drawn according to the correlated task model of Definition 1, $(x_i)_{i=1}^{n+1} \in \mathbb{R}^d$ be i.i.d. with $x_i \sim \mathcal{N}(0, \Sigma)$, and $y_i \sim \mathcal{N}(x_i^\top \beta_i, \sigma^2)$ for all $i \in [n+1]$.*

**Definition 3.** *Let the eigen decompositions of $\Sigma$ and $R$ be denoted by $\Sigma = U\mathrm{diag}(s)U^\top$ and $R = E\mathrm{diag}(\lambda)E^\top$, where $s = [s_1 \ \cdots \ s_d]^\top \in \mathbb{R}_{++}^d$ and $\lambda = [\lambda_1 \ \cdots \ \lambda_n]^\top \in \mathbb{R}_+^n$. Let $s_{\min}$ and $s_{\max}$ denote the smallest and largest eigenvalues of $\Sigma$, respectively. Further, let $\lambda_{\min}$ and $\lambda_{\max}$ denote the smallest and largest nonzero eigenvalues of $R$. Define the effective spectral gap of $\Sigma$ and $R$, respectively, as*

$$\Delta_\Sigma := s_{\max} - s_{\min}, \quad \text{and} \quad \Delta_R := \lambda_{\max} - \lambda_{\min}. \tag{13}$$

**Assumption 1.** *The correlation vector $r$ from (12) lies in the range of $E$, i.e., $r = Ea$ for some nonzero $a = [a_1 \ \cdots \ a_n]^\top \in \mathbb{R}^n$.*

Assumption 1 essentially ensures that $r$ (representing the correlations between in-context tasks) can be expressed as a linear transformation of a vector $a$ with at least one nonzero value. Note that since $r$ lies in the range of the eigenvector matrix $E$, it also lies in the range of $R$ defined in (12). This guarantees that the correlation structure is non-degenerate, meaning that all elements of $r$ are influenced by meaningful correlations. Assumption 1 avoids trivial cases where there are no correlations between tasks. By requiring at least one nonzero element in $a$, the assumption ensures that the tasks are interrelated.

The following theorem characterizes the stationary points $(P, \omega)$ of the WPGD objective (2).

**Theorem 2.** *Consider independent linear data as described in Definition 2. Suppose Assumption 1 on the correlation vector $r$ holds. Define the functions $h_1 : \mathbb{R}_+ \to \mathbb{R}_+$ and $h_2 : [1, \infty) \to \mathbb{R}_+$ as*

$$h_1(\bar{\gamma}) := \left( \sum_{i=1}^n \frac{\lambda_i a_i^2}{(1 + \lambda_i \bar{\gamma})^2} \right) \left( \sum_{i=1}^n \frac{a_i^2}{(1 + \lambda_i \bar{\gamma})^2} \right)^{-1}, \tag{14a}$$

$$h_2(\gamma) := \left( 1 + M \left( \sum_{i=1}^d \frac{s_i^2}{(M + s_i \gamma)^2} \right) \left( \sum_{i=1}^d \frac{s_i^3}{(M + s_i \gamma)^2} \right)^{-1} \right)^{-1}, \tag{14b}$$

where $\{s_i\}_{i=1}^d$ and $\{\lambda_i\}_{i=1}^n$ are the eigenvalues of $\Sigma$ and $R$, respectively; $\{a_i\}_{i=1}^n$ are given in Assumption 1; and $M = \sigma^2 + \sum_{i=1}^d s_i$.

The risk function $\mathcal{L}_{\text{WPGD}}(P, \omega)$ in (2) has a stationary point $(P^\star, \omega^\star)$, up to rescaling, defined as

$$P^\star = \Sigma^{-\frac{1}{2}} \left( \frac{\gamma^\star}{M} \cdot \Sigma + I \right)^{-1} \Sigma^{-\frac{1}{2}}, \quad \text{and} \quad \omega^\star = (h_2(\gamma^\star) \cdot R + I)^{-1} r, \tag{15}$$

where $\gamma^\star$ is a fixed point of composite function $h_1(h_2(\gamma)) + 1$.

Theorem 2 characterizes the stationary points $(P^\star, \omega^\star)$, which exist up to re-scaling. This result presents the first landscape analysis of GLA for the joint learning of $(P, \omega)$, while also exploring the stationary points $(P^\star, \omega^\star)$. In the following, we provide mild conditions on effective spectral gaps of $R$ and $\Sigma$ under which a unique (global) minimum $(P^\star, \omega^\star)$ exists.

**Theorem 3** (Uniqueness of the WPGD Predictor). *Consider independent linear data as given in Definition 2. Suppose Assumption 1 on the correlation vector $r$ holds, and*

$$\Delta_\Sigma \cdot \Delta_R < M + s_{\min}, \tag{16}$$

*where $\Delta_\Sigma$ and $\Delta_R$ denote the effective spectral gaps of $\Sigma$ and $R$, respectively, as given in (13); $s_{\min}$ is the smallest eigenvalue of $\Sigma$; and $M = \sigma^2 + \sum_{i=1}^d s_i$.*

**T1.** *The function $h_1(h_2(\gamma)) + 1$ is a contraction mapping and admits a unique fixed point $\gamma = \gamma^\star$.*

**T2.** *The loss $\mathcal{L}_{\text{WPGD}}(P, \omega)$ has a unique (global) minima $(P^\star, \omega^\star)$, up to re-scaling, given by (15).*

Theorem 3 establishes mild conditions under which a unique (global) minimum $(P^\star, \omega^\star)$ exists, up to scaling invariance, and guarantees the uniqueness of the associated WPGD algorithm. It provides the first global landscape analysis for GLA and generalizes prior work (Li et al., 2024b; Ahn et al., 2024) on the global landscape by extending the optimization properties of linear attention to the more complex GLA with joint $(P, \omega)$ optimization.

**Remark 1.** An interesting observation about the optimal gating parameter $\omega^\star$ is its connection to the correlation matrix $R$, which captures the task correlations in a multitask learning setting. Specifically, the optimal gating given in (15) highlights how $\omega^\star$ depends directly on both the task correlation matrix $R$ and the vector $r$, which encodes the correlations between the tasks and the target task.

**Remark 2.** Condition (16) provides a *sufficient* condition for the uniqueness of a fixed point. This implies that whenever $\Delta_\Sigma \cdot \Delta_R < M + s_{\min}$, the mapping $h_1(h_2(\gamma)) + 1$ is a contraction, ensuring the existence of a unique fixed point. However, there may be cases where the mapping $h_1(h_2(\gamma)) + 1$ does not satisfy Condition (16), yet a unique fixed point (and a unique global minimum) still exists. This is because the Banach Fixed-Point Theorem does not provide a *necessary* condition.

**Corollary 1.** *Suppose $\Sigma = I$. Then, $\Delta_\Sigma = 0$, satisfying Condition (16), and we have $h_2(\gamma^\star) = \frac{1}{d+\sigma^2+1}$, which yields $P^\star = I$ and $\omega^\star = (R + (d + \sigma^2 + 1)I)^{-1} r$. Thus, the optimal risk $\mathcal{L}_{\text{WPGD}}^\star$ defined in (2) is given by*

$$\mathcal{L}_{\text{WPGD}}^\star = d + \sigma^2 - d \cdot r^\top \left( R + (d + \sigma^2 + 1)I \right)^{-1} r. \tag{17}$$

## 5 Optimization landscape of GLA

In this section, we analyze the loss landscape for training a one-layer GLA model and explore the scenarios under which GLA can reach the optimal WPGD risk.

### 5.1 Multi-task prompt model

Following Definitions 1 and 2, we consider the multi-task prompt setting with $K$ correlated tasks $(\beta_k)_{k=1}^K$ and one query task $\beta$. For each correlated task $k \in [K]$, a prompt of length

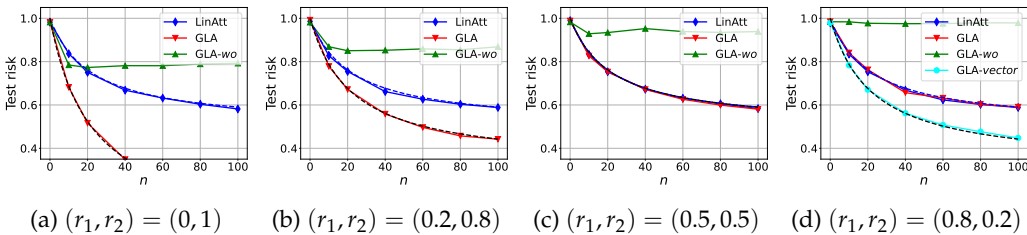

Figure 1: We consider four different types of model training: LinAtt (blue solid): Standard linear attention training. GLA (red solid): GLA training using prompts with delimiters (see (19)) and scalar gating. GLA-wo (green solid): GLA training using prompts without delimiters and with scalar gating. GLA-vector (cyan solid): GLA training using prompts with delimiters and vector gating. The blue and black dashed curves represent the optimal linear attention and WPGD risks from (24) and (17), respectively, as the number of in-context examples $n$ increases. Implementation details are provided in Appendix B.

$n_k$ is drawn, consisting of IID input-label pairs $\{(x_{ik}, y_{ik})\}_{i=1}^{n_k}$ where $y_{ik} \sim \mathcal{N}(\beta_k^\top x_{ik}, \sigma^2)$ to obtain sequence $Z_k = \begin{bmatrix} x_{1k} & \cdots & x_{n_k,k} \\ y_{1k} & \cdots & y_{n_k,k} \end{bmatrix}^\top$. Let $n := \sum_{k=1}^K n_k$.

These sequences $(Z_k)_{k=1}^K$, along with the query token $z_{n+1} = [x_{n+1} \ 0]^\top$, are concatenated to form a single prompt $Z$. Recall (GLA), and let $f_{\mathsf{GLA}}(Z)$ denote the GLA prediction as defined in Theorem 1. Additionally, consider the model construction in (9), where $P_q, P_k \in \mathbb{R}^{d \times d}$ are trainable parameters. The GLA optimization problem is described as follows:

$$\mathcal{L}_{\mathsf{GLA}}^\star := \min_{P_k, P_q \in \mathbb{R}^{d \times d}, G \in \mathcal{G}} \mathcal{L}_{\mathsf{GLA}}(P_k, P_q, G),$$

$$\text{where} \quad \mathcal{L}_{\mathsf{GLA}}(P_k, P_q, G) = \mathbb{E}\left[(y - f_{\mathsf{GLA}}(Z))^2\right]. \tag{18}$$

Here, $G(\cdot)$ represents the gating function and $\mathcal{G}$ denotes the function search space, which is determined by the chosen gating strategies (cf. Table 1 in Yang et al. (2023)).

Note that 1) the task vectors $(\beta_k)_{k=1}^K$ are not explicitly shown in the prompt, 2) in-context features $x_{ik}$ are randomly drawn, and 3) the gating function is applied to the tokens $(Z_k)_{k=1}^K$. Given the above three evidences, the implicit weighting induced by the GLA model varies across different prompts, and it prevents the GLA from learning the optimal weighting. To address this, we introduce delimiters to mark the boundary of each task. Let $(d_k)_{k=1}^K$ be the delimiters that determine stop of the tasks. Specifically, the final prompt is given by

$$Z = \begin{bmatrix} Z_1^\top & d_1 & \cdots & Z_K^\top & d_K & z_{n+1} \end{bmatrix}^\top. \tag{19}$$

Additionally, to decouple the influence of gating and data, we envision that each token is $z_i = [x_i^\top \ y_i \ c_i^\top]^\top$ where $c_i \neq 0 \in \mathbb{R}^p$ is the contextual features with $p$ being its dimension and $(x_i, y_i)$ are the data features. Let $\bar{c}_0, \cdots, \bar{c}_K \in \mathbb{R}^p$ be $K+1$ contextual feature vectors.

- For task prompts $Z_k$: Contextual features are set to a fixed vector $\bar{c}_0 \neq 0$.

- For delimiters $d_k$: Data features (e.g., $x_i, y_i$) are set to zero so that $d_k = [0_{d+1}^\top \ \bar{c}_k^\top]^\top$.

Explicit delimiters have been utilized in addressing real-world problems (Wang et al., 2024a; Asai et al., 2022; Dun et al., 2023) due to their ability to improve efficiency and enhance generalization, particularly in task-mixture or multi-document scenarios. To further motivate the introduction of $(d_k)_{k=1}^K$, we present in Figure 1 the results of GLA training with and without delimiters, represented by the red and green curves, respectively. The black dashed curves indicate the optimal WPGD loss, $\mathcal{L}_{\mathsf{WPGD}}^\star$, under different scenarios. Notably, training GLA without delimiters (the green solid curve) performs strictly worse. In contrast, training with delimiters can achieve optimal performance under certain conditions (see Figures 1a, 1b, and 1c). Theorem 4, presented in the next section, provides a theoretical explanation for these observations, including the misalignment observed in Figure 1d.

## 5.2 Loss landscape of one-layer GLA

Given the input tokens with extended dimension, to ensure that GLA still implements WPGD as in Theorem 1, we propose the following model construction.

$$\tilde{W}_k = \begin{bmatrix} W_k & 0 \\ 0 & 0 \end{bmatrix}, \quad \tilde{W}_q = \begin{bmatrix} W_q & 0 \\ 0 & 0 \end{bmatrix}, \quad \text{and} \quad \tilde{W}_v = \begin{bmatrix} W_v & 0 \\ 0 & 0 \end{bmatrix}. \tag{20}$$

Here, $\tilde{W}_{k,q,v} \in \mathbb{R}^{(d+p+1)\times(d+p+1)}$ and $W_{k,q,v} \in \mathbb{R}^{(d+1)\times(d+1)}$ are constructed via (9). The main idea is to set the last $p$ rows and columns of attention matrices to zeros, ensuring that the delimiters do not affect the final prediction.

**Assumption 2.** *Contextual feature vectors $\bar{c}_0, \cdots, \bar{c}_K$ are linearly independent, and activation function $\phi(z) : \mathbb{R} \to [0,1]$ is continuous, satisfying $\phi(-\infty) = 0$ and $\phi(+\infty) = 1$.*

**Assumption 3.** *The correlation between context tasks $(\beta_k)_{k=1}^K$ and query task $\beta$ satisfies $\mathbb{E}[\beta_i^\top \beta_j] = 0$ and $\mathbb{E}[\beta_i^\top \beta] \leq \mathbb{E}[\beta_j^\top \beta]$ for all $1 \leq i \leq j \leq K$.*

Given context examples $\{(X_k, y_k) := (x_{ik}, y_{ik})_{i=1}^{n_k}\}_{k=1}^K$, define the concatenated data $(X, y)$:

$$X = \begin{bmatrix} X_1^\top & \cdots & X_K^\top \end{bmatrix}^\top \in \mathbb{R}^{n\times d}, \quad \text{and} \quad y = \begin{bmatrix} y_1^\top & \cdots & y_K^\top \end{bmatrix}^\top \in \mathbb{R}^n. \tag{21}$$

Based on the assumptions above, we are able to establish the equivalence between optimizing one-layer GLA and optimizing one-step WPGD predictor under scalar gating.

**Theorem 4** (Scalar Gating). *Suppose Assumption 2 holds. Recap the function $\mathcal{L}_{\text{WPGD}}(P, \omega)$ from (2) with dataset $(X, y)$ defined in (21). Consider (GLA) with input prompt $Z$ defined in (19), the model constructions described in (20), and scalar gating $G(z) = \phi(w_g^\top z)\mathbf{1}_{(d+p+1)\times(d+p+1)}$, where $w_g$ is a trainable parameter. Then, the optimal risk $\mathcal{L}_{\text{GLA}}^\star$ defined in (18) obeys*

$$\mathcal{L}_{\text{GLA}}^\star = \mathcal{L}_{\text{WPGD}}^{\star,\mathcal{W}}, \quad \text{where} \quad \mathcal{L}_{\text{WPGD}}^{\star,\mathcal{W}} := \min_{P \in \mathbb{R}^{d\times d}, \omega \in \mathcal{W}} \mathcal{L}_{\text{WPGD}}(P, \omega). \tag{22}$$

*Here, $\mathcal{W} := \left\{ \begin{bmatrix} \omega_1 \mathbf{1}_{n_1}^\top & \cdots & \omega_K \mathbf{1}_{n_K}^\top \end{bmatrix}^\top \in \mathbb{R}^n \;\middle|\; 0 \leq \omega_i \leq \omega_j \leq 1, \forall 1 \leq i \leq j \leq K \right\}.$*

*Additionally, suppose Assumption 3 holds and $n_i = n_j$, for all $i, j \in [K]$. Let $\mathcal{L}_{\text{WPGD}}^\star$ be the optimal WPGD risk (cf. (2)). Then $\mathcal{L}_{\text{GLA}}^\star$ satisfies*

$$\mathcal{L}_{\text{GLA}}^\star = \mathcal{L}_{\text{WPGD}}^\star. \tag{23}$$

Assumption 2 ensures that any $\omega$ in $\mathcal{W}$ can be achieved by an appropriate choice of gating parameters. Furthermore, Assumption 3 guarantees that the optimal choice of $\omega$ under the WPGD objective lies within the search space $\mathcal{W}$. The proof is provided in Appendix F.1.

In Figure 1, we conduct model training to validate our findings. Consider the setting where $K = 2$ and let $(r_1, r_2) = \left(\mathbb{E}[\beta_1^\top \beta]/d, \mathbb{E}[\beta_2^\top \beta]/d\right)$. In Figures 1a, 1b, and 1c, Assumption 3 holds, and the GLA results (shown in solid red) align with the optimal WPGD risk (represented by the dashed black), validating (23). However, in Figure 1d, since $r_1 > r_2$, Assumption 3 does not hold, and as a result, the optimal GLA loss $\mathcal{L}_{\text{GLA}}^\star$ obtained from (22) is lower than the optimal WPGD loss $\mathcal{L}_{\text{WPGD}}^\star$. Further details are deferred to Appendix B.

**Loss landscape of vector gating.** Till now, our discussion has focused on the scalar gating setting. It is important to highlight that, even in the scalar-weighting context, analyzing the WPGD problem remains non-trivial due to the joint optimization over $(P, \omega)$. However, as demonstrated in Theorem 4, scalar gating can only express weightings within the set $\mathcal{W}$. If Assumption 3 does not hold, $\mathcal{L}_{\text{GLA}}^\star$ cannot achieve the optimal WPGD loss (see the misalignment between red solid curve, presenting $\mathcal{L}_{\text{GLA}}^\star$, and black dashed curve, presenting $\mathcal{L}_{\text{WPGD}}^\star$ in Figure 1d). We argue that vector gating overcomes this limitation by applying distinct weighting mechanisms across different dimensions, facilitating stronger expressivity.

**Theorem 5** (Vector Gating). *Suppose Assumption 2 holds. Consider (GLA) with input prompt $Z$ from (19), the model constructions from (20) but with $\tilde{W}_v = \begin{bmatrix} \mathbf{0}_{(d+p+1)\times d} & u & \mathbf{0}_{(d+p+1)\times p} \end{bmatrix}^\top$,*

and a vector gating $G(z) = \phi(\boldsymbol{W}_g z)\boldsymbol{1}_{d+p+1}^\top$. *Recap Problem* (18) *with prediction defined as* $f_{\mathsf{GLA}}(\boldsymbol{Z}) := \boldsymbol{o}_{n+1}^\top \boldsymbol{h}$, *where* $\boldsymbol{h} \in \mathbb{R}^{d+p+1}$ *is a linear prediction head. Here,* $\boldsymbol{u}$, $\boldsymbol{W}_g$ *and* $\boldsymbol{h}$ *are trainable parameters. Let* $\mathcal{L}_{\mathsf{GLA}\text{-}v}^\star$ *denote its optimal risk, and* $\mathcal{L}_{\mathsf{WPGD}}^\star$ *be defined as in* (2). *Then,* $\mathcal{L}_{\mathsf{GLA}\text{-}v}^\star = \mathcal{L}_{\mathsf{WPGD}}^\star$.

In Theorem 4, the equivalence between $\mathcal{L}_{\mathsf{GLA}}^\star$ and $\mathcal{L}_{\mathsf{WPGD}}^\star$ is established only when both Assumptions 2 and 3 are satisfied. In contrast, Theorem 5 demonstrates that applying vector gating requires only Assumption 2 to establish $\mathcal{L}_{\mathsf{GLA}\text{-}v}^\star = \mathcal{L}_{\mathsf{WPGD}}^\star$. Specifically, under the bounded activation model of Assumption 2, scalar gating is unable to express non-monotonic weighting schemes. For instance, suppose there are two tasks: Even if Task 1 is more relevant to the query, Assumption 2 will assign a higher (or equal) weight to examples in Task 2 resulting in sub-optimal prediction. Theorem 5 shows that vector gating can avoid such bottlenecks by potentially encoding tasks in distinct subspaces. To verify these intuitions, in Figure 1d, we train a GLA model with vector gating and results are presented in cyan curve, which outperform the scalar gating results (red solid) and align with the optimal WPGD loss (black dashed).

**Loss landscape of one-layer linear attention.** Given the fact that linear attention is equivalent to (GLA) when implemented with all ones gating, that is, $\boldsymbol{G}_i \equiv \boldsymbol{1}$, we derive the following corollary. Consider training a standard single-layer linear attention, i.e., by setting $\boldsymbol{G}_i \equiv \boldsymbol{1}$ in (GLA), and let $f_{\mathsf{ATT}}(\boldsymbol{Z})$ be its prediction. Let $\mathcal{L}_{\mathsf{ATT}}^\star$ be the corresponding optimal risk following (18).

**Corollary 2.** *Consider a single-layer linear attention following model construction in* (9) *and let linear data as given in Definition 2. Let* $\boldsymbol{R}$ *and* $\boldsymbol{r}$ *be the corresponding correlation matrix and vector as defined in Definition 1. Suppose* $\boldsymbol{\Sigma} = \boldsymbol{I}$. *Then, the optimal risk obeys*

$$\mathcal{L}_{\mathsf{ATT}}^\star := \min_{\boldsymbol{P} \in \mathbb{R}^{d \times d}} \mathcal{L}_{\mathsf{WPGD}}(\boldsymbol{P}, \boldsymbol{\omega} = \boldsymbol{1}) = d + \sigma^2 - \frac{d(\boldsymbol{1}^\top \boldsymbol{r})^2}{n(d + \sigma^2 + 1) + \boldsymbol{1}^\top \boldsymbol{R}\boldsymbol{1}}. \tag{24}$$

In the Figure 1, blue solid curves represent the linear attention results and blue dashed are the theory curves following (24). The two curves are aligned in all the subfigures, which validate our Corollary 2. More implementation details are deferred to Appendix B.

## 6 Discussion

Our analysis is currently limited to GLA models under specific weight construction assumptions as outlined in Section 3, which may not capture the full expressivity of practical GLA implementations. The theoretical framework we developed requires certain structural constraints on the attention matrices (as specified in Equation 9) to establish the connection with WPGD algorithms.

Several important directions for future research include:

1. Extending our analysis to more general GLA architectures without the weight construction constraints, which would better reflect deployed models such as Mamba (Gu & Dao, 2023; Dao & Gu, 2024) and RWKV (Peng et al., 2024).

2. Investigating when delimiters are necessary for effective learning in multi-task settings. Our experiments show that without delimiters, GLA models perform sub-optimally, but a deeper theoretical characterization of this phenomenon is needed.

3. Exploring the GLA landscape where gating functions depend on input features in more complex ways than our current formulation allows.

4. Analyzing the effects of incorporating MLP layers between GLA layers, which could enhance the model's expressive power beyond what we have characterized.

Addressing these limitations would further bridge the gap between our theoretical understanding and the practical performance of state-of-the-art GLA-based sequence models.

## Acknowledgements

Y.L. and S.O. were supported in part by the NSF grants CCF2046816, CCF-2403075, CCF-2008020, the Office of Naval Research grant N000142412289, and by gifts/awards from Open Philanthropy, OpenAI, Amazon Research, and Google Research. M.F. was supported in part by awards NSF TRIPODS II 2023166, CCF 2007036, CCF 2212261, and CCF 2312775.

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

## A  Capabilities of multi-layer GLA

Ahn et al. (2024) demonstrated that, with appropriate construction, an $L$-layer linear attention model performs $L$-step preconditioned GD on the dataset $(x_i, y_i)_{i=1}^n$ provided within the prompt. In this work, we study multi-layer GLA and analyze the associated algorithm class it can emulate. It is worth mentioning that Ahn et al. (2024) does not consider *causal masking* which is integral to multilayer GLA due to its recurrent nature described in (GLA). Our analysis will capture the impact of gating and causal mask through $n$ separate GD trajectories that are coupled.

**Multi-layer GLA.** For any input prompt $Z \in \mathbb{R}^{(n+1)\times(d+1)}$, let $\mathsf{GLA}(Z) :=$ $\begin{bmatrix} o_1 & o_2 & \cdots & o_{n+1} \end{bmatrix}^\top \in \mathbb{R}^{(n+1)\times(d+1)}$ denote its GLA output, as defined in (GLA). We consider an $L$-layer GLA model as follows: For each layer $\ell \in [L]$, let $Z_\ell$ denote its input, where we set $Z_1 = Z$, and let $W_{k,\ell}, W_{q,\ell}, W_{v,\ell}$ denote the key, query, and value matrices for layer $\ell$, respectively. After applying a residual connection, the input of the $\ell$-th layer is given by

$$Z_\ell := Z_{\ell-1} + \mathsf{GLA}_{\ell-1}(Z_{\ell-1}). \tag{25}$$

Here, $\mathsf{GLA}_\ell(\cdot)$ denotes the output of the $\ell$-th GLA layer, associated with the attention matrices $(W_{q,\ell}, W_{k,\ell}, W_{v,\ell})$ for all $\ell \in [L]$. In the following, we focus on the $(d+1)$-th entry of each token's output at each layer (after the residual connection), denoted by $o_{i,\ell} := [Z_\ell + \mathsf{GLA}_\ell(Z_\ell)]_{i,d+1}$ for all $i \in [n+1]$ and $\ell \in [L]$.

**Theorem 6.** *Consider an $L$-layer GLA defined in (25), where $W_{k,\ell}, W_{q,\ell}$ in the $\ell$'th layer parameterized by $P_{k,\ell}, -P_{q,\ell} \in \mathbb{R}^{d\times d}$, for all $\ell \in [L]$ following (9). Let the gating be a function of the features, e.g., $G_i = G(x_i)$, and define $\Omega$ as in Theorem 1. Additionally, denote the masking as $M_i = \begin{bmatrix} I_i & 0 \\ 0 & 0 \end{bmatrix} \in \mathbb{R}^{n\times n}$, and let $\hat{\beta}_0 = \beta_{i,0} = 0$ for all $i \in [n]$. Recall that $o_{i,\ell}$ for all $i \in [n+1]$ and $\ell \in [L]$ stands for the last entry of the $i$'th token output at the $\ell$'th layer. We have*

- *For all $i \le n$, $o_{i,\ell} = y_i - x_i^\top \beta_{i,\ell}$, where $\beta_{i,\ell} = \beta_{i,\ell-1} - P_{q,\ell}(\nabla_{i,\ell} \oslash g_{i:n+1})$;*

- *$o_{n+1,\ell} = -x^\top \hat{\beta}_\ell$, where $\hat{\beta}_\ell = (1-\alpha_\ell)\hat{\beta}_{\ell-1} - P_{q,\ell}(\nabla_{n,\ell} \oslash g_{n+1})$, and $\alpha_\ell = x^\top P_{q,\ell} P_{k,\ell}^\top x$.*

*Here, letting $B_\ell = [\beta_{1,\ell} \cdots \beta_{n,\ell}]^\top$, $X_\ell = XP_{k,\ell} \odot \Omega$, and $y_\ell = \mathrm{diag}(XB_\ell^\top)$, we define*

$$\nabla_{i,\ell} = X_\ell^\top M_i (y_\ell - y).$$

We defer the proof of Theorem 6 to the Appendix D.2. Theorem 6 states that $L$-layer GLA (25) implements $L$ steps of WPGD with gradient in a recurrent form and additional weight decay $\alpha_\ell$. To recap, given data $(X, y)$ and prediction $\hat{\beta}$, the gradient with respect to the squared loss takes the form $X^\top(X\hat{\beta} - y)$, up to some constant $c$. In comparison, $P_{q,\ell}(\nabla_{i,\ell} \oslash g_{i:n+1})$ similarly acts as a gradient but incorporates layer-wise feature preconditioners $(P_{q,\ell}, P_{k,\ell})$, data weighting $(\Omega)$, and causality $(g_{i:n+1}, M_i)$. Here, $M_i$ represents causal masking, ensuring that at time $i$, only inputs from $j \le i$ are used for prediction. Notably, the recurrent structure of GLA allows the gating mechanism to apply context-dependent weighting strategies.

To simplify the theorem statement, we assume that the gating function depends only on the input feature, e.g., $G_i = G(x_i)$, ensuring that the corresponding data-dependent weighting is uniform across all layers. This assumption is included solely for clarity in the theorem statement, and if the gating function varies across layers or depends on the entire $(d+1)$-dimensional token rather than just the feature $x_i$, each layer has its own gating $\Omega_\ell$ and $X_\ell = XP_{k,\ell} \odot \Omega_\ell$. Note that our inclusion of the additional term $\alpha_\ell$ captures the influence of the last token's output on the next layer's prediction. Based on the above $L$-layer GLA result, we have the following corollary for multi-layer linear attention network with causal mask in each layer.

**Corollary 3.** *Consider an L-layer linear attention model with causal mask and residual connection in each layer. Let $\ell$'th layer follows the weight construction in (9) with $\boldsymbol{W}_{k,\ell}, \boldsymbol{W}_{q,\ell}$ parameterized by $\boldsymbol{P}_{k,\ell}, -\boldsymbol{P}_{q,\ell}$ as in Theorem 6 and define $\boldsymbol{P}_\ell := \boldsymbol{P}_{q,\ell}\boldsymbol{P}_{k,\ell}^\top$, for $\ell \in [L]$. Let $\hat{\boldsymbol{\beta}}_0 = \mathbf{0}$. Then, the $(d+1)$'th entry of the last token output of each layer satisfies:*

$$o_{n+1,\ell} = -\boldsymbol{x}^\top \hat{\boldsymbol{\beta}}_\ell, \quad \text{where} \quad \hat{\boldsymbol{\beta}}_\ell = (1 - \alpha_\ell)\hat{\boldsymbol{\beta}}_{\ell-1} - \boldsymbol{P}_\ell \boldsymbol{X}^\top (\boldsymbol{y}_\ell - \boldsymbol{y}).$$

*Here, we define $\alpha_\ell = \boldsymbol{x}^\top \boldsymbol{P}_\ell \boldsymbol{x}$ and $\boldsymbol{y}_\ell$ follows the same definition as in Theorem 6.*

Note that Corollary 3 generalizes (Ahn et al., 2024, Lemma 1) by extending it to causal attention. When considering full attention (i.e., without applying the causal mask), we have $\boldsymbol{y}_\ell = \boldsymbol{X}\hat{\boldsymbol{\beta}}_\ell$, and the expression $\boldsymbol{X}^\top(\boldsymbol{y}_\ell - \boldsymbol{y}) = \boldsymbol{X}^\top(\boldsymbol{X}\hat{\boldsymbol{\beta}}_\ell - \boldsymbol{y})$ corresponds to the standard gradient of the squared loss. Furthermore, if we disregard the impact of the last token (e.g., by incorporating the matrix $M$ as in Eq. (3) of Ahn et al. (2024)), then $\alpha_\ell = 0$. These results are also consistent with Ding et al. (2023), which demonstrate that causal masking limits convergence by introducing sequence biases, akin to online gradient descent with non-decaying step sizes.

Our theoretical results in Theorem 6 focus on multi-layer GLA without Multi-Layer Perceptron (MLP) layers to isolate and analyze the effects of the gating mechanism. However, MLP layers, a key component of standard Transformers, facilitate further nonlinear feature transformations and interactions, potentially enhancing GLA's expressive power. Future work could explore the theoretical foundations of integrating MLPs into GLA and analyze the optimization landscape of general gated attention models, aligning them more closely with conventional Transformer architectures (Gu & Dao, 2023; Dao & Gu, 2024; Peng et al., 2024).

# B Experimental setup

## B.1 Implementation detail

**Data generation.** Consider ICL problem with input in the form of multi-task prompt as described in Section 5.1. In the experiments, we set $K = 2$, dimensions $d = 10$ and $p = 5$, uniform context length $n_1 = n_2 = \bar{n}$ where we have $n = K\bar{n}$, and vary $\bar{n}$ from 0 to 50. Let $(r_1, r_2) := \left(\mathbb{E}[\boldsymbol{\beta}_1^\top \boldsymbol{\beta}]/d, \mathbb{E}[\boldsymbol{\beta}_2^\top \boldsymbol{\beta}]/d\right)$ denote the correlations between in-context tasks $\boldsymbol{\beta}_1, \boldsymbol{\beta}_2$ and query task $\boldsymbol{\beta}$. We generate task vectors as follows:

$$\boldsymbol{\beta}_1, \boldsymbol{\beta}_2 \sim \mathcal{N}(0, \boldsymbol{I}_d), \quad \text{and} \quad \boldsymbol{\beta} \sim \mathcal{N}(r_1\boldsymbol{\beta}_1 + r_2\boldsymbol{\beta}_2, (1 - r_1^2 - r_2^2)\boldsymbol{I}_d).$$

Input features are randomly sampled $((\boldsymbol{x}_{ik})_{i=1}^{\bar{n}})_{k=1}^K, \boldsymbol{x}_{n+1} \sim \mathcal{N}(0, \boldsymbol{I}_d)$, and we have $y_{ik} = \boldsymbol{\beta}_k^\top \boldsymbol{x}_{ik}$ ($\sigma = 0$), $k \in \{1, 2\}$ and $y_{n+1} = \boldsymbol{\beta}^\top \boldsymbol{x}_{n+1}$. Additionally, delimiters $\bar{\boldsymbol{c}}_0, \cdots, \bar{\boldsymbol{c}}_K$ are randomly sampled from $\mathcal{N}(0, \boldsymbol{I}_p)$.

**Implementation setting.** We train 1-layer linear attention and GLA models for solving multi-prompt ICL problem as described in Section 5.1. For GLA model, we consider sigmoid-type gating function given by scalar gating: $G(\boldsymbol{z}) = \phi(\boldsymbol{w}_g^\top \boldsymbol{z})\mathbf{1}_{(d+p+1)\times(d+p+1)}$, or vector gating: $G(\boldsymbol{z}) = \phi(\boldsymbol{W}_g \boldsymbol{z})\mathbf{1}_{d+p+1}^\top$ where $\phi(z) = (1 + e^{-z})^{-1}$ is the activation function. Note that although the theoretical results are based on the model constructions (c.f. (9) and (20)), we do not restrict the attention weights in our implementation. We train each model for 10000 iterations with batch size 256 and Adam optimizer with learning rate $10^{-3}$. Similar to the previous work (Li et al., 2024b), since our study focuses on the optimization landscape, ICL problems using linear attention/GLA models are non-convex, and experiments are implemented via gradient descent, we repeat 10 model trainings from different model initialization and data sampling (e.g., different choice of delimiters) and results are presented as the minimal test risk among those 10 trails. Results presented have been normalized by $d$.

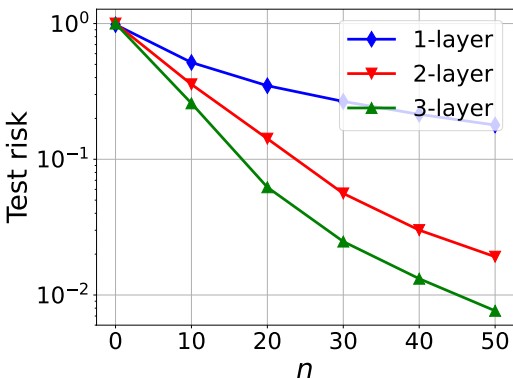

Figure 2: Multi-layer GLA experiments with $(r_1, r_2) = (0, 1)$.

**Experimental results.**  Based on the experimental setting, we can obtain the correlation matrix and vector following Definition 1

$$R = \begin{bmatrix} \mathbf{1}_{\bar{n}} \mathbf{1}_{\bar{n}}^\top & \mathbf{0} \\ \mathbf{0} & \mathbf{1}_{\bar{n}} \mathbf{1}_{\bar{n}}^\top \end{bmatrix} \quad \text{and} \quad r = \begin{bmatrix} r_1 \mathbf{1}_{\bar{n}}^\top & r_2 \mathbf{1}_{\bar{n}}^\top \end{bmatrix}^\top.$$

Then dotted curves display our theoretical results derive using $\Sigma = I$ and $R, r$ above. Specifically, in Figure 1, black dashed curves represent $\mathcal{L}_{\mathsf{WPGD}}^\star$ following (17) and blues dashed curves represent $\mathcal{L}_{\mathsf{GLA}}^\star$ following (24). We consider scenarios where $(r_1, r_2) \in \{(0, 1), (0.2, 0.8), (0.5, 0.5), (0.8, 0.2)\}$ and results are presented in Figures (1a), (1b), (1c) and (1d), respectively.

- `GLA-wo` achieves the worst performance among all the methods. We claim that it is due to the randomness of input tokens as discussed in Section 5.1. Thanks to the introduction of delimiters as described in (19), data and gating is decoupled and a task-dependent weighting is learnt. Hence, `GLA` is able to achieve comparable performance to the optimal one ($\mathcal{L}_{\mathsf{WPGD}}^\star$, red dashed). Note that `GLA-wo` performs even worse than `LinAtt`. It comes from the fact the weighting induced by `GLA-wo` varies over different input prompts and it can not implement all ones weight.

- The alignments between `LinAtt` (blue solid) and blue dashed curves validate our Corollary 2. In Figures 1a, 1b and 1c, the alignments between `GLA` (red solid) and $\mathcal{L}_{\mathsf{WPGD}}$ (black dashed) verify our Theorem 4, specifically, Equation 23. While in 1c and 1d, `GLA` achieves the same performance as `LinAtt`. It is due to the fact that `GLA` can not weight the history higher than its present. Then the equal-weighting, e.g., $\omega = \mathbf{1}$, is the optimal weighting given such constraint. What's more, the alignment between `GLA-vector` (cyan curves) and red dashed in Figure 1d validates our vector gating theorem in Theorem 5.

### B.2  Multi-layer experiments

In this section, we present additional experiments on multi-layer GLA models. We adopt the same experimental setup as described in Figure 1a and Appendix B, with parameters setting to $(r_1, r_2) = (0, 1)$. The results are displayed in Figure 2, where the blue, red, and green curves correspond to the performance of one-, two-, and three-layer GLA models, respectively, with the $y$-axis presented in log-scale. According to Theorem 6, an $L$-layer GLA performs $L$ steps of WPGD, suggesting that deeper models should yield improved predictive performance. The experimental findings in Figure 2 align with the theoretical predictions of Theorem 6.

## C  Related work

**Efficient sequence models.**  Recent sequence model proposals – such as RetNet (Sun et al., 2023), Mamba (Gu & Dao, 2023), xLSTM (Beck et al., 2024), GLA Transformer (Yang et al.,

2023), RWKV-6 (Peng et al., 2024) – admit efficient recurrent forms while being increasingly competitive with the transformer architecture with softmax-attention. However, we have a rather limited theoretical understanding of these architectures, especially, when it comes to their optimization landscape and ICL capabilities. Park et al. (2024); Grazzi et al. (2024) demonstrate that Mamba is effective and competitive with a transformer of similar size in various ICL tasks, whereas Arora et al. (2024); Jelassi et al. (2024) establish theoretical and empirical shortcomings of recurrent models for solving recall tasks. It is worth mentioning that, GLA models also connect to state-space models and linear RNNs (De et al., 2024; Orvieto et al., 2023; Gu et al., 2021; Fu et al., 2022), as they could be viewed as time-varying SSMs (Dao & Gu, 2024; Sieber et al., 2024). Finally, GLA models are also closely related to implicit self-attention frameworks. For example, the work by Zimerman et al. (2024) on unified implicit attention highlights how models such as Mamba (Gu & Dao, 2023) and RWKV (Peng et al., 2023) can be viewed under a shared attention mechanism. Additionally, Zong et al. (2024) leverage gated cross-attention for robust multimodal fusion, demonstrating another practical application of gated mechanisms. Both approaches align with GLA's data-dependent gating, suggesting its potential for explainability and stable fusion tasks.

**Theory of in-context learning.** The theoretical aspects of ICL has been studied by a growing body of works during the past few years (Xie et al., 2022; von Oswald et al., 2023; Gatmiry et al., 2024; Li et al., 2023b; Collins et al., 2024; Wu et al., 2023; Fu et al., 2023; Lin & Lee, 2024; Akyürek et al., 2023; Zhang et al., 2024). A subset of these follow the setting of Garg et al. (2022) which investigates the ICL ability of transformers by focusing on prompts where each example is labeled by a task function from a specific function class, such as linear models. Akyürek et al. (2023) focuses on linear regression and provide a transformer construction that can perform a single step of GD based on in-context examples. Similarly, Von Oswald et al. (2023) provide a construction of weights in linear attention-only transformers that can replicate GD steps for a linear regression task on in-context examples. Notably, they observe similarities between their constructed networks and those resulting from training on ICL prompts for linear regression tasks. Building on these, Zhang et al. (2024); Mahankali et al. (2023); Ahn et al. (2024) focus on the loss landscape of ICL for linear attention models. For a single-layer model trained on in-context prompts for random linear regression tasks, Mahankali et al. (2023); Ahn et al. (2024) show that the resulting model performs a single preconditioned GD step on in-context examples in a test prompt, aligning with the findings of Von Oswald et al. (2023). In contrast, real-world LLMs incorporate additional components (e.g., softmax attention, causal masking, MLPs) that enable more sophisticated learning and lead to expected performance gaps (Shen et al., 2024). GLA models helped bridge the gap from linear attention to transformers. More recent work (Ding et al., 2023) analyzes the challenges of causal masking in causal language models (causalLM), showing that their suboptimal convergence dynamics closely resemble those of online gradient descent with non-decaying step sizes. Additionally, Li et al. (2024b) analyzes the landscape of the H3 architecture, an SSM, under the same dataset model. They show that H3 can implement WPGD thanks to its convolutional/SSM filter. However, their WPGD theory is restricted to the trivial setting of equal weights, relying on the standard prompt model with IID examples and shared tasks. In contrast, we propose novel multitask datasets and prompt models where nontrivial weighting is provably optimal. This allows us to characterize the loss landscape of WPGD and explore advanced GLA models, linking them to data-dependent WPGD algorithms.

**Optimization landscape of attention mechanisms.** The optimization behavior of attention mechanisms has become a rapidly growing area of research (Deora et al., 2023; Huang et al., 2023; Tian et al., 2023; Fu et al., 2023; Li et al., 2024a; Tarzanagh et al., 2024; 2023; Deng et al., 2023; Makkuva et al., 2024; Jeon et al., 2024; Zheng et al., 2023; Collins et al., 2024; Chen & Li, 2024; Li et al., 2023a; Ildiz et al., 2024; Vasudeva et al., 2024b; Bao et al., 2024; Chen et al., 2024; Huang et al., 2024; Wang et al., 2024b; Sun et al., 2025). Particularly relevant to our work are studies investigating convex relaxation approaches to optimize attention models (Sahiner et al., 2022; Ergen et al., 2022), gradient-based optimization methods in vision transformers (Jelassi et al., 2022), optimization dynamics in prompt-attention (Oymak et al., 2023; Tarzanagh et al., 2024), implicit bias of gradient descent

for attention optimization (Tarzanagh et al., 2024; 2023; Julistiono et al., 2024; Li et al., 2024a; Thrampoulidis, 2024; Zhao et al., 2024; Vasudeva et al., 2024a; Sheen et al., 2024), and optimization geometry analysis of next-token prediction (Zhao & Thrampoulidis, 2025). While these works mainly focus on standard attention, our work provides the first optimization landscape analysis of *gated* attention, establishing conditions for a unique global minimum and providing comprehensive insights into the implications of data-dependent weighting in the optimization dynamics encountered during training.

## D GLA ⇔ WPGD

### D.1 Proof of Theroem 1

*Proof.* Recap the problem settings from Section 2 where in-context samples are given by

$$
Z = [z_1 \cdots z_n \, z_{n+1}]^\top = \begin{bmatrix} x_1 & \cdots & x_n & x_{n+1} \\ y_1 & \cdots & y_n & 0 \end{bmatrix}^\top
$$

and let the value, key and query embeddings at time $i$ be

$$
v_i = W_v^\top z_i, \quad k_i = W_k^\top z_i, \quad \text{and} \quad q_i = W_q^\top z_i.
$$

Then we can rewrite the GLA output (cf. (GLA)) as follows:

$$
\begin{aligned}
o_i = S_i q_i \quad \text{and} \quad S_i &= G_i \odot S_{i-1} + v_i k_i^\top \\
&= G_i \odot \cdots G_1 \odot v_1 k_1^\top + \cdots + G_i \odot v_{i-1} k_{i-1}^\top + v_i k_i^\top \\
&= \sum_{j=1}^{i} G_{j:i} \odot v_j k_j^\top,
\end{aligned}
$$

where we define

$$
G_{j:i} = G_{j+1} \odot G_{j+2} \cdots G_i, \quad j < i, \quad \text{and} \quad G_{i:i} = \mathbf{1}_{(d+1)\times(d+1)}.
$$

Consider the prediction based on the last token, then we obtain

$$
o_{n+1} = S_{n+1} q_{n+1} \quad \text{and} \quad S_{n+1} = \sum_{j=1}^{n+1} G_{j:n+1} \odot v_j k_j^\top.
$$

**Construction 1:** Recall the model construction from (9) where

$$
W_k = \begin{bmatrix} P_k & 0 \\ 0 & 0 \end{bmatrix}, \quad W_q = \begin{bmatrix} P_q & 0 \\ 0 & 0 \end{bmatrix} \quad \text{and} \quad W_v = \begin{bmatrix} 0_{d\times d} & 0 \\ 0 & 1 \end{bmatrix}. \tag{26}
$$

Then, given each token $z_i = [x_i^\top \, y_i]^\top, i \in [n]$, single-layer GLA returns

$$
v_i = \begin{bmatrix} 0 \\ y_i \end{bmatrix}, \quad k_i = \begin{bmatrix} P_k^\top x_i \\ 0 \end{bmatrix}, \quad \text{and} \quad q_i = \begin{bmatrix} P_q^\top x_i \\ 0 \end{bmatrix},
$$

and we obtain

$$
v_i k_i^\top = \begin{bmatrix} 0_{d\times d} & 0 \\ y_i x_i^\top P_k & 0 \end{bmatrix}, \quad i \le n, \quad \text{and} \quad v_{n+1} k_{n+1}^\top = 0_{(d+1)\times(d+1)}.
$$

Therefore, since only $d$ entries in $v_i k_i^\top$ matrix are nonzero, given $\odot$ as the Hadamard product, only the corresponding $d$ entries in all $G_i$ matrices are useful. Based on this observation, let

$$
G_i = \begin{bmatrix} * & * \\ g_i^\top & * \end{bmatrix} \quad \text{and} \quad G_{j:i} = \begin{bmatrix} * & * \\ g_{j:i}^\top & * \end{bmatrix},
$$

where $g_{j:i} = g_{j+1} \odot g_{j+2} \cdots g_i \in \mathbb{R}^d$ for $j < i$ and $g_{i:i} = \mathbf{1}_d$.

Combing all together, and letting $X, y$ follow the definitions in (6), we obtain

$$
\begin{aligned}
o_{n+1} &= S_{n+1} q_{n+1} \\
&= \left( \sum_{j=1}^{n+1} G_{j:n+1} \odot v_j k_j^\top \right) q_{n+1} \\
&= \begin{bmatrix} \mathbf{0}_{d \times d} & \mathbf{0} \\ \sum_{j=1}^{n} y_j x_j^\top P_k \odot g_{j:n+1}^\top & 0 \end{bmatrix} \begin{bmatrix} P_q^\top x \\ 0 \end{bmatrix} \\
&= \begin{bmatrix} \mathbf{0} \\ x^\top P_q \left( X P_k \odot \Omega \right)^\top y \end{bmatrix}
\end{aligned}
$$

where

$$
\Omega = \begin{bmatrix} g_{1:n+1} & g_{2:n+1} & \cdots & g_{n:n+1} \end{bmatrix} \in \mathbb{R}^{n \times d}.
$$

Then if taking the last entry of $o_{n+1}$ as final prediction, we get

$$
f_{\mathsf{GLA}}(Z) = x^\top P_q \left( X P_k \odot \Omega \right)^\top y
$$

which completes the proof of Theorem 1. □

**Construction 2:** Based on the construction given in (26), only $d$ elements of $G_i$ matrices are useful. One might ask about the effect of other entries of $G_i$. Therefore, in the following, we introduce an other model construction showing that different row of $G_i$ implements WPGD with different weighting. Similarly, let $W_k, W_q$ be the same as (26) but with $W_v$ constructed by

$$
W_v = \begin{bmatrix} \mathbf{0}_{(d+1) \times d} & u \end{bmatrix}^\top \quad \text{where} \quad u = [u_1 \, u_2 \, \cdots \, u_{d+1}]^\top \in \mathbb{R}^{d+1}.
$$

Then, the value embeddings have the form of $v_i = y_i u$, which gives

$$
v_i k_i^\top = u \begin{bmatrix} y_i x_i^\top P_k & \mathbf{0} \end{bmatrix}.
$$

Next, let

$$
G_i = \begin{bmatrix} (g_i^1)^\top & * \\ (g_i^2)^\top & * \\ \vdots & \vdots \\ (g_i^{d+1})^\top & * \end{bmatrix}, \quad \text{and} \quad G_{j:i} = \begin{bmatrix} (g_{j:i}^1)^\top & * \\ (g_{j:i}^2)^\top & * \\ \vdots & \vdots \\ (g_{j:i}^{d+1})^\top & * \end{bmatrix},
$$

where $g_i^{i'} \in \mathbb{R}^d$ corresponds to the $i'$-th row of $G_i$ and $g_{j:i}^{i'} = g_{j+1}^{i'} \odot g_{j+1}^{i'} \cdots g_i^{i'}$. Then we get the output

$$
o_{n+1} = \begin{bmatrix} \sum_{j=1}^{n} u_1 y_j x_j^\top P_k \odot (g_{j:n+1}^1)^\top & 0 \\ \sum_{j=1}^{n} u_2 y_j x_j^\top P_k \odot (g_{j:n+1}^2)^\top & 0 \\ \vdots \\ \sum_{j=1}^{n} u_{d+1} y_j x_j^\top P_k \odot (g_{j:n+1}^{d+1})^\top & 0 \end{bmatrix} \begin{bmatrix} P_q^\top x \\ 0 \end{bmatrix} = \begin{bmatrix} x^\top P_q \left( X P_k \odot \Omega_1 \right)^\top y \\ x^\top P_q \left( X P_k \odot \Omega_2 \right)^\top y \\ \vdots \\ x^\top P_q \left( X P_k \odot \Omega_{d+1} \right)^\top y \end{bmatrix},
$$

where

$$
\Omega_i = u_i \begin{bmatrix} g_{1:n+1}^i & g_{2:n+1}^i & \cdots & g_{n:n+1}^i \end{bmatrix} \in \mathbb{R}^{n \times d}, \quad i \leq d+1. \tag{27}
$$

Therefore, consider $(d+1)$-dimensional output $o_{n+1}$. Each entry implements a 1-step WPGD with same preconditioners $P_k, P_q$ and different weighting matrices $\Omega$'s. The weighting matrix of $i$'th entry is determined by the $i$'th row of all gating matrices. Note that if consider the last entry of $o_{n+1}$ as prediction, it returns the same result as **Construction 1** above, where only last rows of $G_i$'s are useful.

Additionally, suppose that the final prediction is given after a linear head $h$, that is, $f_{\mathsf{GLA}}(Z) = h^\top o_{n+1}$, and let $h = [h_1 \, h_2 \, \cdots \, h_{d+1}]^\top \in \mathbb{R}^{d+1}$. Then

$$
f_{\mathsf{GLA}}(Z) = h^\top o_{n+1} = x^\top P_q \left( X P_k \odot \bar{\Omega} \right)^\top y, \tag{28}
$$

where

$$\bar{\mathbf{\Omega}} = \sum_{i=1}^{d+1} h_i \mathbf{\Omega}_i = \sum_{i=1}^{d+1} h_i u_i \begin{bmatrix} \mathbf{g}^i_{1:n+1} & \mathbf{g}^i_{2:n+1} & \cdots & \mathbf{g}^i_{n:n+1} \end{bmatrix} \in \mathbb{R}^{n \times d}. \tag{29}$$

Then, single-layer GLA still returns one-step WPGD with updated weighting matrix.

### D.2 Proof of Theorem 6

In this section, we first prove the following theorem, which differs from Theorem 6 only in that $\mathbf{W}_{q,\ell}$ is parameterized by $\mathbf{P}_{q,\ell}$ instead of $-\mathbf{P}_{q,\ell}$. Theorem 6 can then be directly obtained by setting $\mathbf{P}_{q,\ell}$ to $-\mathbf{P}_{q,\ell}$.

**Theorem 7.** *Consider an L-layer GLA with residual connections, where each layer follows the weight construction specified in* (9). *Let $\mathbf{W}_{k,\ell}$ and $\mathbf{W}_{q,\ell}$ in the $\ell$'th layer parameterized by $\mathbf{P}_{k,\ell}, \mathbf{P}_{q,\ell} \in \mathbb{R}^{d \times d}$, for $\ell \in [L]$. Let the gating be a function of the features, e.g., $\mathbf{G}_i = G(\mathbf{x}_i)$, and define $\mathbf{\Omega}$ as in Theorem 1. Additionally, denote the masking as $\mathbf{M}_i = \begin{bmatrix} \mathbf{I}_i & \mathbf{0} \\ \mathbf{0} & \mathbf{0} \end{bmatrix} \in \mathbb{R}^{n \times n}$, and let $\hat{\boldsymbol{\beta}}_0 = \boldsymbol{\beta}_{i,0} = \mathbf{0}$ for all $i \in [n]$.*

*Then, the last entry of the $i$'th token output at the $\ell$'th layer ($o_{i,\ell}$) is:*

- *For all $i \leq n$, $o_{i,\ell} = y_i - \mathbf{x}_i^\top \boldsymbol{\beta}_{i,\ell}$, where $\boldsymbol{\beta}_{i,\ell} = \boldsymbol{\beta}_{i,\ell-1} + \mathbf{P}_{q,\ell} (\nabla_{i,\ell} \oslash \mathbf{g}_{i:n+1})$,*

- $o_{n+1,\ell} = -\mathbf{x}^\top \hat{\boldsymbol{\beta}}_\ell$, *where* $\hat{\boldsymbol{\beta}}_\ell = (1 + \alpha_\ell)\hat{\boldsymbol{\beta}}_{\ell-1} + \mathbf{P}_{q,\ell} (\nabla_{n,\ell} \oslash \mathbf{g}_{n+1})$, *and* $\alpha_\ell = \mathbf{x}^\top \mathbf{P}_{q,\ell} \mathbf{P}_{k,\ell}^\top \mathbf{x}$.

*Here, letting $\mathbf{B}_\ell = [\boldsymbol{\beta}_{1,\ell} \cdots \boldsymbol{\beta}_{n,\ell}]^\top$, $\mathbf{X}_\ell = \mathbf{X} \mathbf{P}_{k,\ell} \odot \mathbf{\Omega}$, and $\mathbf{y}_\ell = diag(\mathbf{X} \mathbf{B}_\ell^\top)$, we define*

$$\nabla_{i,\ell} = \mathbf{X}_\ell^\top \mathbf{M}_i (\mathbf{y}_\ell - \mathbf{y}).$$

*Proof.* Suppose that the input prompt for $\ell$'th layer is

$$\mathbf{Z}_\ell = \begin{bmatrix} \tilde{\mathbf{x}}_1 & \cdots & \tilde{\mathbf{x}}_n & \tilde{\mathbf{x}}_{n+1} \\ \tilde{y}_1 & \cdots & \tilde{y}_n & \tilde{y}_{n+1} \end{bmatrix}^\top \in \mathbb{R}^{(n+1) \times (d+1)} \tag{30}$$

and let $\mathbf{k}_{i,\ell}, \mathbf{q}_{i,\ell}, \mathbf{v}_{i,\ell}$ be their corresponding key, query and value embeddings. Recap the model construction from (9) and follow the same analysis in Appendix D.1. Let $\mathbf{S}_i^\ell, i \in [n+1]$ be the corresponding states. We have

$$\mathbf{S}_i^\ell = \sum_{j=1}^{i} \mathbf{G}_{j:i}^\ell \odot \mathbf{v}_{j,\ell} \mathbf{k}_{j,\ell}^\top$$

$$= \begin{bmatrix} \mathbf{0}_{d \times d} & \mathbf{0} \\ \sum_{j=1}^{i} \tilde{y}_j \tilde{\mathbf{x}}_j^\top \mathbf{P}_{k,\ell} \odot (\mathbf{g}_{j:i}^\ell)^\top & 0 \end{bmatrix}$$

where

$$\mathbf{G}_i^\ell = G(\tilde{\mathbf{x}}_i) = \begin{bmatrix} * & * \\ (\mathbf{g}_i^\ell)^\top & * \end{bmatrix} \quad \text{and} \quad \mathbf{G}_{j:i}^\ell = \begin{bmatrix} * & * \\ (\mathbf{g}_{j:i}^\ell)^\top & * \end{bmatrix}.$$

Additionally, recap that we have $\mathbf{M}_i = \begin{bmatrix} \mathbf{I}_i & \mathbf{0} \\ \mathbf{0} & \mathbf{0} \end{bmatrix}$ and $\oslash$ denotes Hadamard division. Let

$$\mathbf{\Omega}^\ell = \begin{bmatrix} \mathbf{g}_{i:n+1}^\ell & \cdots & \mathbf{g}_{n:n+1}^\ell \end{bmatrix}.$$

Then, defining

$$\tilde{\mathbf{X}} = [\tilde{\mathbf{x}}_1 \ \tilde{\mathbf{x}}_2 \ \cdots \ \tilde{\mathbf{x}}_n]^\top \in \mathbb{R}^{n \times d} \quad \text{and} \quad \tilde{\mathbf{y}} = [\tilde{y}_1 \ \tilde{y}_2 \ \cdots \ \tilde{y}_n]^\top \in \mathbb{R}^n,$$

and letting $o_{i,\ell}, i \in [n+1]$ be their outputs, we get:

- For $i \leq n$,
$$\sum_{j=1}^{i} \tilde{y}_j \boldsymbol{P}_{k,\ell}^\top \tilde{\boldsymbol{x}}_j \odot \boldsymbol{g}_{j:i}^\ell = \left( \sum_{j=1}^{i} \tilde{y}_j \boldsymbol{P}_{k,\ell}^\top \tilde{\boldsymbol{x}}_j \odot \boldsymbol{g}_{j:n+1}^\ell \right) \oslash \boldsymbol{g}_{i:n+1}^\ell$$
$$= (\tilde{\boldsymbol{X}} \boldsymbol{P}_{k,\ell} \odot \boldsymbol{\Omega}^\ell)^\top \boldsymbol{M}_i \tilde{\boldsymbol{y}} \oslash \boldsymbol{g}_{i:n+1}^\ell,$$

and
$$\boldsymbol{o}_{i,\ell} = \boldsymbol{S}_i^\ell \boldsymbol{q}_{i,\ell} = \begin{bmatrix} \boldsymbol{0}_{d \times d} & \boldsymbol{0} \\ \left( (\tilde{\boldsymbol{X}} \boldsymbol{P}_{k,\ell} \odot \boldsymbol{\Omega}^\ell)^\top \boldsymbol{M}_i \tilde{\boldsymbol{y}} \oslash \boldsymbol{g}_{i:n+1}^\ell \right)^\top & 0 \end{bmatrix} \begin{bmatrix} \boldsymbol{P}_{q,\ell}^\top \tilde{\boldsymbol{x}}_i \\ 0 \end{bmatrix}$$
$$= \begin{bmatrix} \boldsymbol{0} \\ \tilde{\boldsymbol{x}}_i^\top \boldsymbol{P}_{q,\ell} \left( (\tilde{\boldsymbol{X}} \boldsymbol{P}_{k,\ell} \odot \boldsymbol{\Omega}^\ell)^\top \boldsymbol{M}_i \tilde{\boldsymbol{y}} \oslash \boldsymbol{g}_{i:n+1}^\ell \right) \end{bmatrix}. \tag{31}$$

- For $i = n + 1$,
$$\sum_{j=1}^{n+1} \tilde{y}_j \boldsymbol{P}_{k,\ell}^\top \tilde{\boldsymbol{x}}_j \odot \boldsymbol{g}_{j:i}^\ell = (\tilde{\boldsymbol{X}} \boldsymbol{P}_{k,\ell} \odot \boldsymbol{\Omega}^\ell)^\top \tilde{\boldsymbol{y}} + \tilde{y}_{n+1} \boldsymbol{P}_{k,\ell}^\top \tilde{\boldsymbol{x}}_{n+1},$$

and
$$\boldsymbol{o}_{n+1,\ell} = \begin{bmatrix} \boldsymbol{0}_{d \times d} & \boldsymbol{0} \\ \left( (\tilde{\boldsymbol{X}} \boldsymbol{P}_{k,\ell} \odot \boldsymbol{\Omega}^\ell)^\top \tilde{\boldsymbol{y}} + \tilde{y}_{n+1} \boldsymbol{P}_{k,\ell}^\top \tilde{\boldsymbol{x}}_{n+1} \right)^\top & 0 \end{bmatrix} \begin{bmatrix} \boldsymbol{P}_{q,\ell}^\top \tilde{\boldsymbol{x}}_{n+1} \\ 0 \end{bmatrix}$$
$$= \begin{bmatrix} \boldsymbol{0} \\ \tilde{\boldsymbol{x}}_{n+1}^\top \boldsymbol{P}_{q,\ell} \left( (\tilde{\boldsymbol{X}} \boldsymbol{P}_{k,\ell} \odot \boldsymbol{\Omega}^\ell)^\top \tilde{\boldsymbol{y}} + \tilde{y}_{n+1} \boldsymbol{P}_{k,\ell}^\top \tilde{\boldsymbol{x}}_{n+1} \right) \end{bmatrix}$$
$$= \begin{bmatrix} \boldsymbol{0} \\ \tilde{y}_{n+1} \tilde{\boldsymbol{x}}_{n+1}^\top \boldsymbol{P}_{q,\ell} \boldsymbol{P}_{k,\ell}^\top \tilde{\boldsymbol{x}}_{n+1} + \tilde{\boldsymbol{x}}_{n+1}^\top \boldsymbol{P}_{q,\ell} \left( (\tilde{\boldsymbol{X}} \boldsymbol{P}_{k,\ell} \odot \boldsymbol{\Omega}^\ell)^\top \tilde{\boldsymbol{y}} \right). \end{bmatrix} \tag{32}$$

From (31) and (32), the first $d$ entries of all the $(d + 1)$-dimensional outputs (e.g, $\boldsymbol{o}_{i,\ell}$, $i \in [n+1]$) are zero. Given the multi-layer GLA as formulated in (25) where residual connections are applied, we have that the first $d$ dimension of all the input tokens remain the same. That is, $\tilde{\boldsymbol{x}}_i \equiv \boldsymbol{x}_i$. Additionally, since gating only depends on the feature $\boldsymbol{x}_i$, then all the layers return the same gatings (e.g, $\boldsymbol{G}_i^\ell \equiv \boldsymbol{G}_i$).

Note that if the gating function varies across layers or depends on the entire token rather than just the feature $\boldsymbol{x}_i$, each layer will have its own gating, leading to $\boldsymbol{G}_i^\ell \neq \boldsymbol{G}_i$. In this theorem, we assume identical gating matrices across layers for simplicity and clarity. However, an extended version of this theorem, without this assumption, can be directly derived.

Therefore, we obtain
$$\boldsymbol{o}_{i,\ell} = \begin{bmatrix} \boldsymbol{0} \\ \boldsymbol{x}_i^\top \boldsymbol{P}_{q,\ell} \left( \boldsymbol{X}_\ell^\top \boldsymbol{M}_i \tilde{\boldsymbol{y}} \oslash \boldsymbol{g}_{i:n+1} \right) \end{bmatrix}, \quad i \leq n$$
$$\boldsymbol{o}_{n+1,\ell} = \begin{bmatrix} \boldsymbol{0} \\ \tilde{y}_{n+1} \boldsymbol{x}^\top \boldsymbol{P}_{q,\ell} \boldsymbol{P}_{k,\ell}^\top \boldsymbol{x} + \boldsymbol{x}^\top \boldsymbol{P}_{q,\ell} \left( \boldsymbol{X}_\ell^\top \tilde{\boldsymbol{y}} \oslash \boldsymbol{g}_{n+1} \right) \end{bmatrix}.$$
where we define $\boldsymbol{X}_\ell := \boldsymbol{X} \boldsymbol{P}_{k,\ell} \odot \boldsymbol{\Omega}$.

Since we focus on the last/$(d + 1)$'th entry of each token output, we have
$$o_{i,\ell} = \boldsymbol{x}_i^\top \boldsymbol{P}_{q,\ell} \left( \boldsymbol{X}_\ell^\top \boldsymbol{M}_i \tilde{\boldsymbol{y}} \oslash \boldsymbol{g}_{i:n+1} \right), \quad i \leq n,$$
$$o_{n+1,\ell} = \tilde{y}_{n+1} \boldsymbol{x}^\top \boldsymbol{P}_{q,\ell} \boldsymbol{P}_{k,\ell}^\top \boldsymbol{x} + \boldsymbol{x}^\top \boldsymbol{P}_{q,\ell} \left( \boldsymbol{X}_\ell^\top \tilde{\boldsymbol{y}} \oslash \boldsymbol{g}_{n+1} \right). \tag{33}$$

It remains to get the $\tilde{y}_i$ for each layer's input. Let inputs of $(\ell - 1)$'th and $\ell$'th layer be
$$\boldsymbol{Z}_{\ell-1} = \begin{bmatrix} \boldsymbol{x}_1 & \cdots & \boldsymbol{x}_n & \boldsymbol{x} \\ y_1^{\ell-1} & \cdots & y_n^{\ell-1} & y^{\ell-1} \end{bmatrix} \quad \text{and} \quad \boldsymbol{Z}_\ell = \begin{bmatrix} \boldsymbol{x}_1 & \cdots & \boldsymbol{x}_n & \boldsymbol{x} \\ y_1^\ell & \cdots & y_n^\ell & y^\ell \end{bmatrix}.$$
Define
$$\boldsymbol{y}^{\ell-1} := [y_1^{\ell-1} \ \cdots \ y_n^{\ell-1}]^\top \in \mathbb{R}^n.$$

- For $i \leq n$, from (33) and (25), we have

$$y_i^\ell = y_i^{\ell-1} + x_i^\top P_{q,\ell} \left( X_\ell^\top M_i y^{\ell-1} \oslash g_{i:n+1} \right)$$
$$\implies y_i^\ell - y_i^{\ell-1} = x_i^\top P_{q,\ell} \left( X_\ell^\top M_i y^{\ell-1} \oslash g_{i:n+1} \right).$$

Suppose that $y_i - y_i^\ell = x_i^\top \beta_i^\ell$. Then

$$y^\ell = y - [x_1^\top \beta_1^\ell \; \cdots \; x_n^\top \beta_n^\ell]^\top = y - \mathrm{diag}(XB^\top) \tag{34}$$

and

$$y_i - x_i^\top \beta_i^\ell - (y_i - x_i^\top \beta_i^{\ell-1}) = x_i^\top P_{q,\ell} \left( X_\ell^\top M_i (y - \mathrm{diag}(XB^\top)) \oslash g_{i:n+1} \right)$$
$$\implies x_i^\top \beta_i^\ell = x_i^\top \beta_i^{\ell-1} - x_i^\top P_{q,\ell} \left( X_\ell^\top M_i (y - \mathrm{diag}(XB^\top)) \oslash g_{i:n+1} \right).$$

Hence,

$$y_i^\ell = y_i - x_i^\top \beta_i^\ell \quad \text{where} \quad \beta_i^\ell = \beta_i^{\ell-1} - P_{q,\ell} \left( X_\ell^\top M_i (y - \mathrm{diag}(XB^\top)) \oslash g_{i:n+1} \right) \tag{35}$$

- For $i = n + 1$, from (33) and (25), we have

$$y^\ell = y^{\ell-1} + y^{\ell-1} x^\top P_{q,\ell} P_{k,\ell}^\top x + x^\top P_{q,\ell} \left( X_\ell^\top y^{\ell-1} \oslash g_{n+1} \right).$$

Setting $\alpha_\ell = x^\top P_{q,\ell} P_{k,\ell}^\top x$ and recalling $y^\ell$ from (34), we get

$$y^\ell = (1 + \alpha_\ell) y^{\ell-1} + x^\top P_{q,\ell} \left( X_\ell^\top (y - \mathrm{diag}(XB^\top)) \right) \oslash g_{n+1}.$$

Additionally, let $y^\ell = -x^\top \beta^\ell$. Then,

$$-x^\top \beta^\ell = -(1 + \alpha_\ell) x^\top \beta^{\ell-1} + x^\top P_{q,\ell} \left( X_\ell^\top (y - \mathrm{diag}(XB^\top)) \right) \oslash g_{n+1}.$$

Hence,

$$y^\ell = -x^\top \beta^\ell \quad \text{where} \quad \beta^\ell = (1 + \alpha_\ell) \beta^{\ell-1} - P_{q,\ell} \left( X_\ell^\top (y - \mathrm{diag}(XB^\top)) \right) \oslash g_{n+1} \tag{36}$$

Combining (35) and (36) together completes the proof. ☐

## E  Optimization landscape of WPGD

We first provide the following Lemma.

**Lemma 1** (Existence of Fixed Point)**.** *Consider the functions $h_1(\bar{\gamma})$ and $h_2(\gamma)$ defined in (14a) and (14b), respectively. The composite function $h_1(h_2(\gamma)) + 1$ has at least one fixed point $\gamma^\star \geq 1$, i.e., there exists $\gamma^\star \geq 1$ such that $h_1(h_2(\gamma^\star)) + 1 = \gamma^\star$.*

*Proof.* We prove existence using continuity and boundedness arguments along with the intermediate value theorem. Note that $h_1(\bar{\gamma})$ and $h_2(\gamma)$ are continuous functions. Thus, their composition $f(\gamma) := h_1(h_2(\gamma)) + 1$ is continuous for all $\gamma \geq 1$.

We examine the behavior of $f(\gamma)$ at the boundaries of its domain: At $\gamma = 1$, we have $h_2(1) = c_0 > 0$, where $c_0$ is a positive constant. Therefore, $f(1) = h_1(h_2(1)) + 1 = h_1(c_0) + 1 > 0$, which is finite and positive since $h_1$ is positive for all non-negative inputs. As $\gamma \to \infty$, $h_2(\gamma) \to c_\infty > 0$, where $c_\infty$ is a positive constant.

For sufficiently large $\gamma$, we can establish that $f(\gamma) < \gamma$. To see this, note that we have shown that $h_2(\gamma) \to c_\infty$ as $\gamma \to \infty$, where $c_\infty$ is a fixed positive constant. Further, for any fixed positive value $\bar{\gamma}$, the function $h_1(\bar{\gamma})$ is bounded, i.e., $h_1(\bar{\gamma}) \leq \max_{1 \leq i \leq n} \lambda_i$, which follows

from the fact that $h_1(\bar{\gamma})$ is a weighted average of the $\lambda_i$ values. Therefore, there exists a constant $K$ such that $f(\gamma) \leq K$ for all $\gamma \geq 1$. Hence, it follows that for all $\gamma > K$, we have $f(\gamma) \leq K < \gamma$.

Now, consider the function $\bar{f}(\gamma) = f(\gamma) - \gamma$, which is continuous on $[1, \infty)$ since both $f(\gamma)$ and $\gamma$ are continuous. We have $\bar{f}(1) = f(1) - 1 = h_1(c_0) > 0$ and for some $\gamma_1 > K$, we have $\bar{f}(\gamma_1) = f(\gamma_1) - \gamma_1 < 0$. Since $\bar{f}$ is continuous and changes sign from positive to negative on the interval $[1, \gamma_1]$, by the Intermediate Value Theorem, there exists at least one $\gamma^\star \in (1, \gamma_1)$ such that $\bar{f}(\gamma^\star) = 0$. This implies $f(\gamma^\star) - \gamma^\star = 0$, or equivalently, $f(\gamma^\star) = \gamma^\star$. Therefore, there exists at least one fixed point $\gamma^\star \geq 1$ for the function $h_1(h_2(\gamma)) + 1$. □

### E.1 Proof of Theorem 2

*Proof.* Recapping the objective from (2) and following Definitions 1 and 2, we have

$$\mathcal{L}(\boldsymbol{P}, \boldsymbol{\omega}) = \mathbb{E}\left[\left(y - \boldsymbol{x}^\top \boldsymbol{P} \boldsymbol{X}(\boldsymbol{\omega} \odot \boldsymbol{y})\right)^2\right]$$

$$= \mathbb{E}\left[y^2\right] - 2\mathbb{E}\left[y\boldsymbol{x}^\top \boldsymbol{P} \boldsymbol{X}(\boldsymbol{\omega} \odot \boldsymbol{y})\right] + \mathbb{E}\left[\left(\boldsymbol{x}^\top \boldsymbol{P} \boldsymbol{X}(\boldsymbol{\omega} \odot \boldsymbol{y})\right)^2\right].$$

Let $y = \boldsymbol{x}^\top \boldsymbol{\beta} + \xi$ and $y_i = \boldsymbol{x}_i^\top \boldsymbol{\beta}_i + \xi_i$, for all $i \in [n]$, where $\xi, \xi_i \sim \mathcal{N}(0, \sigma^2)$ are i.i.d. Then,

$$\mathbb{E}[y^2] = \mathbb{E}[(\boldsymbol{x}^\top \boldsymbol{\beta} + \xi)^2] = \text{tr}\left(\boldsymbol{\Sigma}\right) + \sigma^2,$$

and

$$\mathbb{E}\left[y\boldsymbol{x}^\top \boldsymbol{P} \boldsymbol{X}(\boldsymbol{\omega} \odot \boldsymbol{y})\right] = \mathbb{E}\left[(\boldsymbol{\beta}^\top \boldsymbol{x} + \xi)\boldsymbol{x}^\top \boldsymbol{P} \sum_{i=1}^n \omega_i \boldsymbol{x}_i(\boldsymbol{x}_i^\top \boldsymbol{\beta}_i + \xi_i)\right]$$

$$= \mathbb{E}\left[\boldsymbol{\beta}^\top \boldsymbol{x}\boldsymbol{x}^\top \boldsymbol{P} \sum_{i=1}^n \omega_i \boldsymbol{x}_i \boldsymbol{x}_i^\top \boldsymbol{\beta}_i\right]$$

$$= \text{tr}\left(\boldsymbol{\Sigma} \boldsymbol{P} \boldsymbol{\Sigma} \sum_{i=1}^n \omega_i \mathbb{E}\left[\boldsymbol{\beta}_i \boldsymbol{\beta}^\top\right]\right)$$

$$= \text{tr}\left(\boldsymbol{\Sigma}^2 \boldsymbol{P}\right) \boldsymbol{\omega}^\top \boldsymbol{r}.$$

Here, the last equality comes from the fact that since $\boldsymbol{\beta}_i - r_{ij}\boldsymbol{\beta}_j$ is independent of $\boldsymbol{\beta}_j$ for $i, j \in [n+1]$ following Definition 1, we have $\mathbb{E}\left[\boldsymbol{\beta}_i \boldsymbol{\beta}^\top\right] = r_{i,n+1}\boldsymbol{I}_d$ and $\sum_{i=1}^n \omega_i \mathbb{E}\left[\boldsymbol{\beta}_i \boldsymbol{\beta}^\top\right]$ returns $\boldsymbol{\omega}^\top \boldsymbol{r} \cdot \boldsymbol{I}_d$.

Hence,

$$\mathbb{E}\left[\left(\boldsymbol{x}^\top \boldsymbol{P} \boldsymbol{X}(\boldsymbol{\omega} \odot \boldsymbol{y})\right)^2\right] = \mathbb{E}\left[\boldsymbol{x}^\top \boldsymbol{P}\left(\sum_{i=1}^n \omega_i(\boldsymbol{x}_i^\top \boldsymbol{\beta}_i + \xi_i)\boldsymbol{x}_i\right)\left(\sum_{i=1}^n \omega_i \boldsymbol{x}_i^\top(\boldsymbol{x}_i^\top \boldsymbol{\beta}_i + \xi_i)\right)\boldsymbol{P}^\top \boldsymbol{x}\right]$$

$$= \text{tr}\left(\boldsymbol{P}^\top \boldsymbol{\Sigma} \boldsymbol{P} \mathbb{E}\left[\sum_{i=1}^n \omega_i^2(\boldsymbol{x}_i^\top \boldsymbol{\beta}_i + \xi_i)^2 \boldsymbol{x}_i \boldsymbol{x}_i^\top\right]\right)$$

$$+ \text{tr}\left(\boldsymbol{P}^\top \boldsymbol{\Sigma} \boldsymbol{P} \mathbb{E}\left[\sum_{i \neq j} \omega_i \omega_j(\boldsymbol{x}_i^\top \boldsymbol{\beta}_i + \xi_i)\boldsymbol{x}_i \boldsymbol{x}_j^\top(\boldsymbol{x}_j^\top \boldsymbol{\beta}_j + \xi_j)\right]\right),$$

where

$$\text{tr}\left(\boldsymbol{P}^\top \boldsymbol{\Sigma} \boldsymbol{P} \mathbb{E}\left[\sum_{i=1}^n \omega_i^2(\boldsymbol{x}_i^\top \boldsymbol{\beta}_i + \xi_i)^2 \boldsymbol{x}_i \boldsymbol{x}_i^\top\right]\right) = \text{tr}\left(\boldsymbol{P}^\top \boldsymbol{\Sigma} \boldsymbol{P} \mathbb{E}\left[\sum_{i=1}^n \omega_i^2(\boldsymbol{x}_i^\top \boldsymbol{\beta}_i \boldsymbol{\beta}_i^\top \boldsymbol{x}_i + \sigma^2)\boldsymbol{x}_i \boldsymbol{x}_i^\top\right]\right)$$

$$= \|\boldsymbol{\omega}\|_{\ell_2}^2 \, \text{tr}\left(\boldsymbol{P}^\top \boldsymbol{\Sigma} \boldsymbol{P}\left(\mathbb{E}\left[\boldsymbol{x}\boldsymbol{x}^\top \boldsymbol{x}\boldsymbol{x}^\top\right] + \sigma^2 \boldsymbol{\Sigma}\right)\right)$$

$$= \|\boldsymbol{\omega}\|_{\ell_2}^2 \left(\text{tr}\left(\boldsymbol{\Sigma} \boldsymbol{P}^\top \boldsymbol{\Sigma} \boldsymbol{P}\right)\left(\text{tr}\left(\boldsymbol{\Sigma}\right) + \sigma^2\right) + 2\text{tr}\left(\boldsymbol{\Sigma}^2 \boldsymbol{P}^\top \boldsymbol{\Sigma} \boldsymbol{P}\right)\right),$$

and

$$\text{tr}\left(\boldsymbol{P}^\top \boldsymbol{\Sigma} \boldsymbol{P} \mathbb{E}\left[\sum_{i\neq j}\omega_i\omega_j(\boldsymbol{x}_i^\top \boldsymbol{\beta}_i + \xi_i)\boldsymbol{x}_i \boldsymbol{x}_j^\top (\boldsymbol{x}_j^\top \boldsymbol{\beta}_j + \xi_j)\right]\right) = \text{tr}\left(\boldsymbol{P}^\top \boldsymbol{\Sigma} \boldsymbol{P} \mathbb{E}\left[\sum_{i\neq j}\omega_i\omega_j\boldsymbol{x}_i\boldsymbol{x}_i^\top \boldsymbol{\beta}_i\boldsymbol{\beta}_j^\top \boldsymbol{x}_j\boldsymbol{x}_j^\top\right]\right)$$

$$= \text{tr}\left(\boldsymbol{P}^\top \boldsymbol{\Sigma} \boldsymbol{P}\right)\sum_{i\neq j}\omega_i\omega_j\mathbb{E}[\boldsymbol{\beta}_i^\top \boldsymbol{\beta}_j]$$

$$= \text{tr}\left(\boldsymbol{\Sigma}^2 \boldsymbol{P}^\top \boldsymbol{\Sigma} \boldsymbol{P}\right)\boldsymbol{\omega}^\top (\boldsymbol{R} - \boldsymbol{I})\boldsymbol{\omega}.$$

Combining all together and letting $M := \text{tr}\left(\boldsymbol{\Sigma}\right) + \sigma^2$, we obtain

$$\mathcal{L}(\boldsymbol{P},\boldsymbol{\omega}) = M - 2\text{tr}\left(\boldsymbol{\Sigma}^2 \boldsymbol{P}\right)\boldsymbol{\omega}^\top \boldsymbol{r}$$
$$+ M\|\boldsymbol{\omega}\|_{\ell_2}^2 \text{tr}\left(\boldsymbol{\Sigma} \boldsymbol{P}^\top \boldsymbol{\Sigma} \boldsymbol{P}\right) + (\|\boldsymbol{\omega}\|_{\ell_2}^2 + \boldsymbol{\omega}^\top \boldsymbol{R}\boldsymbol{\omega})\text{tr}\left(\boldsymbol{\Sigma}^2 \boldsymbol{P}^\top \boldsymbol{\Sigma} \boldsymbol{P}\right). \tag{37}$$

For simplicity, and without loss of generality, let

$$\check{\boldsymbol{P}} = \sqrt{\boldsymbol{\Sigma}} \boldsymbol{P} \sqrt{\boldsymbol{\Sigma}}. \tag{38}$$

Then, we obtain

$$\mathcal{L}(\check{\boldsymbol{P}},\boldsymbol{\omega}) = M - 2\text{tr}\left(\boldsymbol{\Sigma}\check{\boldsymbol{P}}\right)\boldsymbol{\omega}^\top \boldsymbol{r}$$
$$+ M\|\boldsymbol{\omega}\|_{\ell_2}^2 \text{tr}\left(\check{\boldsymbol{P}}^\top \check{\boldsymbol{P}}\right) + (\|\boldsymbol{\omega}\|_{\ell_2}^2 + \boldsymbol{\omega}^\top \boldsymbol{R}\boldsymbol{\omega})\text{tr}\left(\boldsymbol{\Sigma}\check{\boldsymbol{P}}^\top \check{\boldsymbol{P}}\right). \tag{39}$$

Further, the gradients can be written as

$$\nabla_{\check{\boldsymbol{P}}}\mathcal{L}(\check{\boldsymbol{P}},\boldsymbol{\omega}) = -2\boldsymbol{\omega}^\top \boldsymbol{r}\boldsymbol{\Sigma} + 2M\|\boldsymbol{\omega}\|_{\ell_2}^2 \check{\boldsymbol{P}} + 2(\|\boldsymbol{\omega}\|_{\ell_2}^2 + \boldsymbol{\omega}^\top \boldsymbol{R}\boldsymbol{\omega})\boldsymbol{\Sigma}\check{\boldsymbol{P}}, \tag{40}$$

$$\nabla_{\boldsymbol{\omega}}\mathcal{L}(\check{\boldsymbol{P}},\boldsymbol{\omega}) = -2\text{tr}\left(\boldsymbol{\Sigma}\check{\boldsymbol{P}}\right)\boldsymbol{r} + 2M\text{tr}\left(\check{\boldsymbol{P}}^\top \check{\boldsymbol{P}}\right)\boldsymbol{\omega} + 2\text{tr}\left(\boldsymbol{\Sigma}\check{\boldsymbol{P}}^\top \check{\boldsymbol{P}}\right)(\boldsymbol{I}_n + \boldsymbol{R})\boldsymbol{\omega}. \tag{41}$$

Using the first-order optimality condition, and setting $\nabla_{\check{\boldsymbol{P}}}\mathcal{L}(\check{\boldsymbol{P}},\boldsymbol{\omega}) = 0$ and $\nabla_{\boldsymbol{\omega}}\mathcal{L}(\check{\boldsymbol{P}},\boldsymbol{\omega}) = 0$, we obtain

$$\check{\boldsymbol{P}} = \left(M\|\boldsymbol{\omega}\|_{\ell_2}^2 \boldsymbol{I} + (\|\boldsymbol{\omega}\|_{\ell_2}^2 + \boldsymbol{\omega}^\top \boldsymbol{R}\boldsymbol{\omega})\boldsymbol{\Sigma}\right)^{-1}\boldsymbol{\Sigma}\boldsymbol{\omega}^\top \boldsymbol{r}$$

$$= \frac{\boldsymbol{\omega}^\top \boldsymbol{r}}{M\|\boldsymbol{\omega}\|_{\ell_2}^2}\left(\frac{\|\boldsymbol{\omega}\|_{\ell_2}^2 + \boldsymbol{\omega}^\top \boldsymbol{R}\boldsymbol{\omega}}{M\|\boldsymbol{\omega}\|_{\ell_2}^2}\boldsymbol{I} + \boldsymbol{\Sigma}^{-1}\right)^{-1} \tag{42a}$$

$$= \frac{\boldsymbol{\omega}^\top \boldsymbol{r}}{M\|\boldsymbol{\omega}\|_{\ell_2}^2}\left(\frac{\gamma}{M}\cdot \boldsymbol{I} + \boldsymbol{\Sigma}^{-1}\right)^{-1},$$

where

$$\gamma := \frac{\boldsymbol{\omega}^\top \boldsymbol{R}\boldsymbol{\omega}}{\|\boldsymbol{\omega}\|_{\ell_2}^2} + 1.$$

Further,

$$\boldsymbol{\omega} = \left(\left(M\text{tr}\left(\check{\boldsymbol{P}}^\top \check{\boldsymbol{P}}\right) + \text{tr}\left(\boldsymbol{\Sigma}\check{\boldsymbol{P}}^\top \check{\boldsymbol{P}}\right)\right)\boldsymbol{I} + \text{tr}\left(\boldsymbol{\Sigma}\check{\boldsymbol{P}}^\top \check{\boldsymbol{P}}\right)\boldsymbol{R}\right)^{-1}\text{tr}\left(\boldsymbol{\Sigma}\check{\boldsymbol{P}}\right)\boldsymbol{r}$$

$$= \frac{\text{tr}\left(\boldsymbol{\Sigma}\check{\boldsymbol{P}}\right)}{M\text{tr}\left(\check{\boldsymbol{P}}^\top \check{\boldsymbol{P}}\right) + \text{tr}\left(\boldsymbol{\Sigma}\check{\boldsymbol{P}}^\top \check{\boldsymbol{P}}\right)}\left(\boldsymbol{I} + \frac{\text{tr}\left(\boldsymbol{\Sigma}\check{\boldsymbol{P}}^\top \check{\boldsymbol{P}}\right)}{M\text{tr}\left(\check{\boldsymbol{P}}^\top \check{\boldsymbol{P}}\right) + \text{tr}\left(\boldsymbol{\Sigma}\check{\boldsymbol{P}}^\top \check{\boldsymbol{P}}\right)}\boldsymbol{R}\right)^{-1}\boldsymbol{r}. \tag{42b}$$

Let

$$\boldsymbol{\Sigma}_\gamma := \frac{\gamma}{M}\cdot \boldsymbol{I} + \boldsymbol{\Sigma}^{-1}.$$

Then, we get

$$
\begin{aligned}
\frac{\mathrm{tr}\left(\Sigma\tilde{P}^\top\tilde{P}\right)}{M\mathrm{tr}\left(\tilde{P}^\top\tilde{P}\right)+\mathrm{tr}\left(\Sigma\tilde{P}^\top\tilde{P}\right)} &= \left(1+M\frac{\mathrm{tr}\left(\tilde{P}^\top\tilde{P}\right)}{\mathrm{tr}\left(\Sigma\tilde{P}^\top\tilde{P}\right)}\right)^{-1} \\
&= \left(1+M\frac{\mathrm{tr}\left(\Sigma_\gamma^{-2}\right)}{\mathrm{tr}\left(\Sigma\Sigma_\gamma^{-2}\right)}\right)^{-1} \\
&= \left(1+M\sum_{i=1}^{d}\frac{s_i^2}{(M+\gamma s_i)^2}\left(\sum_{i=1}^{d}\frac{s_i^3}{(M+\gamma s_i)^2}\right)^{-1}\right)^{-1} \\
&=: h_2(\gamma).
\end{aligned}
$$

Here, the last equality follows from eigen decomposition $\Sigma = U\mathrm{diag}(s)U^\top$ with $s = [s_1,\ldots,s_d]^\top \in \mathbb{R}_{++}^d$.

Now, plugging $\tilde{P}$ defined in (42a) within $\omega$ given in (42b), we obtain

$$
\omega = \frac{\mathrm{tr}\left(\Sigma\tilde{P}\right)}{M\mathrm{tr}\left(\tilde{P}^\top\tilde{P}\right)+\mathrm{tr}\left(\Sigma\tilde{P}^\top\tilde{P}\right)}\cdot\left(h_2(\gamma)\cdot R+I\right)^{-1}r. \tag{43}
$$

Using the above formulae for $\omega$, we rewrite $\gamma = \omega^\top R\omega/\|\omega\|_{\ell_2}^2 + 1$ as

$$
\begin{aligned}
\gamma - 1 &= \frac{r^\top(h_2(\gamma)R+I)^{-1}R(h_2(\gamma)R+I)^{-1}r}{r^\top(h_2(\gamma)R+I)^{-2}r} \\
&= \sum_{i=1}^{n}\frac{\lambda_i a_i^2}{(1+h_2(\gamma)\lambda_i)^2}\left(\sum_{i=1}^{n}\frac{a_i^2}{(1+h_2(\gamma)\lambda_i)^2}\right)^{-1} \\
&=: h_1(h_2(\gamma)),
\end{aligned} \tag{44}
$$

where the second equality follows from Assumption 1 where $r = Ea$ with $a = [a_1,\cdots,a_n]^\top \in \mathbb{R}^n$, and the fact that $R = E\mathrm{diag}(\lambda)E^\top$ denotes the eigen decomposition of $R$, with $\lambda = [\lambda_1,\ldots,\lambda_n]^\top \in \mathbb{R}_+^n$.

It follows from Lemma 1 that there exists $\gamma^\star \geq 1$ such that $h_1(h_2(\gamma^\star)) + 1 = \gamma^\star$. From (42a) and (43), we obtain

$$
\begin{aligned}
\tilde{P} &= C(r,\omega,\Sigma)\cdot\left(\frac{\gamma^\star}{M}\cdot I+\Sigma^{-1}\right)^{-1}, \quad\text{and} \\
\omega &= c(r,\omega,\Sigma)\cdot\left(h_2(\gamma^\star)\cdot R+I\right)^{-1}r.
\end{aligned} \tag{45}
$$

for some $C(r,\omega,\Sigma) = \frac{\omega^\top r}{M\|\omega\|_{\ell_2}^2}$ and $c(r,\omega,\Sigma) = \frac{\mathrm{tr}(\Sigma\tilde{P})}{M\mathrm{tr}(\tilde{P}^\top\tilde{P})+\mathrm{tr}(\Sigma\tilde{P}^\top\tilde{P})}$.

Now, using the our definition $\tilde{P} = \sqrt{\Sigma}P\sqrt{\Sigma}$, we obtain

$$
\begin{aligned}
P(\gamma^\star) &= C(r,\omega,\Sigma)\cdot\Sigma^{-\frac{1}{2}}\left(\frac{\gamma^\star}{M}\cdot\Sigma+I\right)^{-1}\Sigma^{-\frac{1}{2}}, \quad\text{and} \\
\omega(\gamma^\star) &= c(r,\omega,\Sigma)\cdot\left(h_2(\gamma^\star)\cdot R+I\right)^{-1}r.
\end{aligned}
$$

This completes the proof. $\qquad\square$

### E.2 Proof of Theorem 3

We first provide the following Lemma.

**Lemma 2.** *Consider the functions $h_1(\bar{\gamma})$ and $h_2(\gamma)$ defined in (14a) and (14b), respectively, where $\lambda_i \geq 0$, $a_i \neq 0$ for at least one $i$, $s_i > 0$, and $M = \sigma^2 + \sum_{i=1}^d s_i > 0$. Suppose $\Delta_\Sigma \cdot \Delta_R < M + s_{\min}$, where $\Delta_\Sigma$ and $\Delta_R$ denote the effective spectral gaps of $\Sigma$ and $R$, respectively, as given in (13); and $s_{\min}$ is the smallest eigenvalue of $\Sigma$. We have that*

$$\left| \frac{\partial h_2}{\partial \gamma} \cdot \frac{\partial h_1}{\partial h_2} \right| \leq \frac{\Delta_\Sigma^2 \cdot \Delta_R^2}{(M + s_{\min})^2} < 1.$$

*Proof.* Let

$$B(\gamma) = \sum_{i=1}^d \frac{s_i^3}{(M + \gamma s_i)^2}, \quad C(\gamma) = \sum_{i=1}^d \frac{s_i^2}{(M + \gamma s_i)^2}, \quad A(\gamma) = 1 + M\frac{C(\gamma)}{B(\gamma)}.$$

The derivatives of $B(\gamma)$ and $C(\gamma)$ are

$$B'(\gamma) = -2\sum_{i=1}^d \frac{s_i^4}{(M + \gamma s_i)^3}, \quad C'(\gamma) = -2\sum_{i=1}^d \frac{s_i^3}{(M + \gamma s_i)^3}.$$

The gradient of $h_2(\gamma) = A(\gamma)^{-1}$ is

$$\frac{\partial h_2}{\partial \gamma} = -M \left( \frac{1}{A(\gamma)B(\gamma)} \right)^2 \left( C'(\gamma)B(\gamma) - C(\gamma)B'(\gamma) \right). \tag{46}$$

It can be seen that

$$\left( \frac{1}{A(\gamma)} \right)^2 \leq M^{-2} \left( \sum_{i=1}^d \frac{s_i^3}{(M + \gamma s_i)^2} \right)^2 \left( \sum_{i=1}^d \frac{s_i^2}{(M + \gamma s_i)^2} \right)^{-2},$$

which implies that

$$\left( \frac{1}{A(\gamma)B(\gamma)} \right)^2 \leq M^{-2} \left( \sum_{i=1}^d \frac{s_i^2}{(M + \gamma s_i)^2} \right)^{-2}. \tag{47a}$$

Further, we have

$$
\begin{aligned}
C'(\gamma)B(\gamma) - C(\gamma)B'(\gamma) &= -2\sum_{i=1}^d \frac{s_i^3}{(M + \gamma s_i)^3} \sum_{i=1}^d \frac{s_i^3}{(M + \gamma s_i)^2} \\
&\quad + 2\sum_{i=1}^d \frac{s_i^2}{(M + \gamma s_i)^2} \sum_{i=1}^d \frac{s_i^4}{(M + \gamma s_i)^3} \\
&= \sum_{i=1}^d \sum_{j=1}^d \frac{T_{ij}}{(M + \gamma s_i)^3 (M + \gamma s_j)^3} \\
&= M \cdot \sum_{i=1}^d \sum_{j=1}^d \frac{s_i^2 s_j^2 (s_i - s_i)^2}{(M + \gamma s_i)^3 (M + \gamma s_j)^3},
\end{aligned}
\tag{47b}
$$

where

$$
\begin{aligned}
T_{ij} &= s_i^2(M + \gamma s_i)s_j^4 + s_i^4 s_j^2(M + \gamma s_j) \\
&\quad - s_i^3 s_j^3(M + \gamma s_j) - s_i^3(M + \gamma s_i)s_j^3 \\
&= s_i^2 s_j^2 \left( M \cdot (s_j^2 + s_i^2 - 2s_i s_j) + \gamma(s_i s_j^2 + s_i^2 s_j - s_i s_j^2 - s_i^2 s_j) \right).
\end{aligned}
\tag{47c}
$$

Thus, substituting (47a) and (47b) into (46), we obtain

$$\left| \frac{\partial h_2}{\partial \gamma} \right| \leq M \cdot M^{-1} \left( \sum_{i=1}^d \frac{s_i^2}{(M + \gamma s_i)^2} \right)^{-2} \sum_{i,j=1}^d \frac{s_i^2 s_j^2 (s_i - s_j)^2}{(M + \gamma s_i)^3 (M + \gamma s_j)^3}. \tag{48}$$

Next, we derive $\frac{\partial h_1}{\partial \bar{\gamma}}$. Let

$$D(\bar{\gamma}) = \sum_{i=1}^{n} \frac{\lambda_i a_i^2}{(1 + \bar{\gamma}\lambda_i)^2}, \quad E(\bar{\gamma}) = \sum_{i=1}^{n} \frac{a_i^2}{(1 + \bar{\gamma}\lambda_i)^2}.$$

We have

$$D'(\bar{\gamma}) = -2\sum_{i=1}^{n} \frac{\lambda_i^2 a_i^2}{(1 + \bar{\gamma}\lambda_i)^3}, \quad E'(\bar{\gamma}) = -2\sum_{i=1}^{n} \frac{\lambda_i a_i^2}{(1 + \bar{\gamma}\lambda_i)^3}.$$

The derivative of $h_1$ with respect to $\bar{\gamma}$ is given by

$$\frac{\partial h_1}{\partial \bar{\gamma}} = -\left(\frac{1}{E(\bar{\gamma})}\right)^2 \left(E(\bar{\gamma})D'(\bar{\gamma}) - D(\bar{\gamma})E'(\bar{\gamma})\right). \tag{49}$$

Substituting into (49), we get

$$\left(\frac{1}{E(\bar{\gamma})}\right)^2 = \left(\sum_{i=1}^{n} \frac{a_i^2}{(1 + \bar{\gamma}\lambda_i)^2}\right)^{-2}, \tag{50a}$$

and

$$\begin{aligned}
E(\bar{\gamma})D'(\bar{\gamma}) - D(\bar{\gamma})E'(\bar{\gamma}) &= 2\sum_{i=1}^{n} \frac{\lambda_i^2 a_i^2}{(1 + \bar{\gamma}\lambda_i)^3} \sum_{i=1}^{n} \frac{a_i^2}{(1 + \bar{\gamma}\lambda_i)^2} \\
&\quad - 2\sum_{i=1}^{n} \frac{\lambda_i a_i^2}{(1 + \bar{\gamma}\lambda_i)^2} \sum_{i=1}^{n} \frac{a_i^2 \lambda_i}{(1 + \bar{\gamma}\lambda_i)^3} \\
&= \sum_{i=1}^{n}\sum_{j=1}^{n} \frac{\bar{T}_{ij}}{(1 + \bar{\gamma}\lambda_i)^3(1 + \bar{\gamma}\lambda_j)^3} \\
&= \sum_{i=1}^{n}\sum_{j=1}^{n} \frac{a_i^2 a_j^2 \left(\lambda_i^2 + \lambda_j^2 - 2\lambda_i\lambda_j\right)}{(1 + \bar{\gamma}\lambda_i)^3(1 + \bar{\gamma}\lambda_j)^3}.
\end{aligned} \tag{50b}$$

Here,

$$\begin{aligned}
\bar{T}_{ij} &= \lambda_i^2 a_i^2 a_j^2 (1 + \bar{\gamma}\lambda_j) + a_i^2(1 + \bar{\gamma}\lambda_i)\lambda_j^2 a_j^2 \\
&\quad - \lambda_i a_i^2(1 + \bar{\gamma}\lambda_i)a_j^2\lambda_j - a_i^2\lambda_i\lambda_j a_j^2(1 + \bar{\gamma}\lambda_j) \\
&= a_i^2 a_j^2 \left(\lambda_i^2(1 + \bar{\gamma}\lambda_j) + (1 + \bar{\gamma}\lambda_i)\lambda_j^2 - \lambda_i(1 + \bar{\gamma}\lambda_i)\lambda_j - \lambda_i\lambda_j(1 + \bar{\gamma}\lambda_j)\right).
\end{aligned} \tag{50c}$$

Hence, substituting (50a) and (50b) into (49) gives

$$\frac{\partial h_1}{\partial \bar{\gamma}} = -\left(\sum_{i=1}^{n} \frac{a_i^2}{(1 + \bar{\gamma}\lambda_i)^2}\right)^{-2} \sum_{i,j=1}^{n} \frac{a_i^2 a_j^2(\lambda_i - \lambda_j)^2}{(1 + \bar{\gamma}\lambda_i)^3(1 + \bar{\gamma}\lambda_j)^3}. \tag{51}$$

Now, for the combined derivative, we have

$$\begin{aligned}
\left|\frac{\partial h_2}{\partial \gamma} \cdot \frac{\partial h_1}{\partial \bar{\gamma}}\right| &\leq \left(\sum_{i=1}^{d} \frac{s_i^2}{(M + \gamma s_i)^2}\right)^{-2} \sum_{i,j=1}^{d} \frac{s_i^2 s_j^2 (s_i - s_j)^2}{(M + \gamma s_i)^3 (M + \gamma s_j)^3} \\
&\quad \cdot \left(\sum_{i=1}^{n} \frac{a_i^2}{(1 + \bar{\gamma}\lambda_i)^2}\right)^{-2} \sum_{i,j=1}^{n} \frac{a_i^2 a_j^2(\lambda_i - \lambda_j)^2}{(1 + \bar{\gamma}\lambda_i)^3(1 + \bar{\gamma}\lambda_j)^3}.
\end{aligned}$$

Note that $M + \gamma s_i$ and $1 + \bar{\gamma}\lambda_j$ are nonnegative for all $i, j$. Hence,

$$\left| \frac{\partial h_2}{\partial \gamma} \cdot \frac{\partial h_1}{\partial \bar{\gamma}} \right| \leq \left( \sum_{i=1}^{d} \frac{s_i^2 \left(M + \gamma s_i\right)}{\left(M + \gamma s_i\right)^3} \right)^{-2}$$
$$\cdot \sum_{i,j=1}^{d} \frac{s_i^2 s_j^2 \cdot \Delta_1 \cdot \left(M + \gamma s_j\right)\left(M + \gamma s_i\right)}{\left(M + \gamma s_i\right)^3 \left(M + \gamma s_j\right)^3}$$
$$\cdot \left( \sum_{i=1}^{n} \frac{a_i^2 (1 + \bar{\gamma}\lambda_i)}{(1 + \bar{\gamma}\lambda_i)^3} \right)^{-2}$$
$$\cdot \sum_{i,j=1}^{d} \frac{a_i^2 a_j^2 \cdot \Delta_2 \cdot (1 + \bar{\gamma}\lambda_i)\left(1 + \bar{\gamma}\lambda_j\right)}{(1 + \bar{\gamma}\lambda_i)^3 \left(1 + \bar{\gamma}\lambda_j\right)^3},$$

where

$$\Delta_1 := \max_{i,j \in \mathcal{S}} \frac{(s_i - s_j)^2}{\left(M + \gamma s_j\right)\left(M + \gamma s_i\right)}, \quad \Delta_2 := \max_{i,j \in \mathcal{S}} \frac{(\lambda_i - \lambda_j)^2}{(1 + \bar{\gamma}\lambda_i)(1 + \bar{\gamma}\lambda_j)}. \tag{52}$$

Here, $\mathcal{S} = \{i \in [n] \mid \lambda_i \neq 0\} \subset [n]$.

Finally, setting $\bar{\gamma} = h_2(\gamma)$, since $\gamma \geq 1$ and by our assumption $\Delta_{\Sigma} \cdot \Delta_R < M + s_{\min}$, we obtain

$$|h_1'(h_2(\gamma)) \cdot h_2'(\gamma)| = \left| \frac{\partial h_2}{\partial \gamma} \cdot \frac{\partial h_1}{\partial h_2} \right| \leq \Delta_1 \cdot \Delta_2 \leq \frac{\Delta_{\Sigma}^2 \cdot \Delta_R^2}{(M + s_{\min})^2} < 1.$$

where $\Delta_{\Sigma}$ and $\Delta_R$ are the spectral gaps of $\Sigma$ and $R$; and $s_{\min}$ is the smallest eigenvalue of $\Sigma$; and $M = \sigma^2 + \sum_{i=1}^{d} s_i$. $\qquad \square$

*Proof of Theorem 3.* It follows from Lemma 1 that there exists $\gamma^\star \geq 1$ such that $h_1(h_2(\gamma^\star)) + 1 = \gamma^\star$. Further, Lemma 2 establishes that the mapping $h_1(h_2(\gamma)) + 1$ is a contraction mapping, since it satisfies $|\partial h_1(h_2(\gamma))/\partial \gamma| < 1$. Therefore, by the Banach fixed-point theorem, there exists a *unique* fixed-point solution, denoted by $\gamma^\star$, satisfying $\gamma^\star = h_1(h_2(\gamma^\star)) + 1$. This completes the proof of statement **T1.**.

Next, we establish **T2.**. First, from Theorem 2, the stationary point $(P^\star, \omega^\star)$, up to rescaling, is defined as

$$P^\star = \Sigma^{-\frac{1}{2}} \left( \frac{\gamma^\star}{M} \cdot \Sigma + I \right)^{-1} \Sigma^{-\frac{1}{2}} \quad \text{and} \quad \omega^\star = (h_2(\gamma^\star) \cdot R + I)^{-1} r,$$

where $\gamma^\star$ is a fixed point of composite function $h_1(h_2(\gamma)) + 1$.

Given the coercive nature of the loss function $\mathcal{L}(P, \omega)$, a global minimum exists. Since the global minimum must satisfy first-order optimality conditions, it must lie within the set of stationary points characterized by Theorem 2.

Now, consider arbitrary global minimizers $(\hat{P}, \hat{\omega})$. Define scaling factors $\alpha, \beta > 0$ and write $(\hat{P}, \hat{\omega}) = (\alpha P^\star, \beta \omega^\star)$. Substituting this scaling into the loss function gives:

$$\mathcal{L}(\alpha P^\star, \beta \omega^\star) = M - 2\alpha\beta \mathrm{tr}\left(\Sigma^2 P^\star\right) \omega^{\star\top} r + M\alpha^2\beta^2 \|\omega^\star\|^2 \mathrm{tr}\left(\Sigma P^{\star\top} \Sigma P^\star\right)$$
$$+ \alpha^2\beta^2 (\|\omega^\star\|^2 + \omega^{\star\top} R\omega^\star) \mathrm{tr}\left(\Sigma^2 P^{\star\top} \Sigma P^\star\right). \tag{53}$$

Differentiating this expression with respect to $\alpha$ and $\beta$ and setting the derivatives equal to zero, we obtain the equation

$$-2A + 2\alpha\beta(MCB + (C + D)E) = 0, \tag{54a}$$

where we define

$$A := \mathrm{tr}\left(\Sigma^2 P^\star\right) \omega^{\star\top} r, \quad B := \mathrm{tr}\left(\Sigma P^{\star\top} \Sigma P^\star\right),$$
$$C := \|\omega^\star\|^2, \quad D := \omega^{\star\top} R\omega^\star, \quad E := \mathrm{tr}\left(\Sigma^2 P^{\star\top} \Sigma P^\star\right). \tag{54b}$$

Equation (54) implies a unique constraint on the product $\alpha\beta$, ensuring no independent rescaling beyond a fixed ratio can further minimize the loss. Hence, no distinct minimizers exist other than scaling the original $(P^\star, \omega^\star)$ pair. Thus, the stationary point $(P^\star, \omega^\star)$ from Theorem 2 is indeed a unique minimizer up to rescaling.

$\square$

### E.3 Proof of Corollary 1

*Proof.* Since by assumption $\Sigma = I$, it follows from (14b) that

$$h_2(\gamma^\star) = \left(1 + (d + \sigma^2) \sum_{i=1}^d \frac{1}{(d + \sigma^2 + \gamma^\star + 1)^2} \left(\sum_{i=1}^d \frac{1}{(d + \sigma^2 + \gamma^\star + 1)^2}\right)^{-1}\right)^{-1}$$

$$= \frac{1}{d + \sigma^2 + 1}.$$

Substituting this into (15) gives

$$P^\star = I, \quad \text{and} \quad \omega^\star = \left(R + (d + \sigma^2 + 1)I\right)^{-1} r.$$

Now, recall that the objective function is given by

$$\mathcal{L}(\omega) = M - 2\mathrm{tr}\left(\Sigma^2 P\right) \omega^\top r + M \|\omega\|_{\ell_2}^2 \, \mathrm{tr}\left(\Sigma P^\top \Sigma P\right) + (\|\omega\|_{\ell_2}^2 + \omega^\top R\omega)\mathrm{tr}\left(\Sigma^2 P^\top \Sigma P\right),$$

and, by assumption, $M = \sigma^2 + d$.

Substituting $P^\star = I$ and $\omega^\star = \left(R + (d + \sigma^2 + 1)I\right)^{-1} r$ into the objective (37), and using $\Sigma = I$, we get:

$$\mathcal{L}(\omega^\star) = (\sigma^2 + d) - 2 \cdot d \cdot r^\top \omega^\star + (\sigma^2 + d) \cdot \|\omega^\star\|^2 d + d \left(\|\omega^\star\|^2 + \omega^{\star\top} R\omega^\star\right).$$

The expression simplifies as

$$\mathcal{L}(\omega^\star) = (\sigma^2 + d) - 2d \cdot r^\top \left(R + (d + \sigma^2 + 1)I\right)^{-1} r + (\sigma^2 + d)d\|\omega^\star\|^2 + d \left(\|\omega^\star\|^2 + \omega^{\star\top} R\omega^\star\right).$$

Next, we compute $\|\omega^\star\|^2$ and $\omega^{\star\top} R\omega^\star$. By definition, we have

$$\|\omega^\star\|^2 = r^\top \left(R + (d + \sigma^2 + 1)I\right)^{-2} r,$$

and

$$\omega^{\star\top} R\omega^\star = r^\top \left(R + (d + \sigma^2 + 1)I\right)^{-1} R \left(R + (d + \sigma^2 + 1)I\right)^{-1} r.$$

Thus,

$$(d + \sigma^2 + 1)\|\omega^\star\|^2 + \omega^{\star\top} R\omega^\star = r^\top \left(R + (d + \sigma^2 + 1)I\right)^{-1}$$

$$\cdot \left((d + \sigma^2 + 1)I + R\right) \left(R + (d + \sigma^2 + 1)I\right)^{-1} r$$

$$= r^\top \left(R + (d + \sigma^2 + 1)I\right)^{-1} r.$$

Substituting this result back into the objective function gives

$$\mathcal{L}(\omega^\star) = (\sigma^2 + d) - d \cdot r^\top \left(R + (d + \sigma^2 + 1)I\right)^{-1} r.$$

$\square$

# F  Loss landscape of 1-layer GLA

## F.1  Proof of Theorem 4

*Proof.* We first prove that under Assumption 2, $\mathcal{L}_{\mathsf{GLA}}^{\star} = \min_{P \in \mathbb{R}^{d \times d}, \omega \in \mathcal{W}} \mathcal{L}_{\mathsf{WPGD}}(P, \omega)$ where $\mathcal{W}$ is the search space of weighting vector $\omega \in \mathbb{R}^n$ defined as

$$\mathcal{W} := \left\{ \left[ \omega_1 \mathbf{1}_{n_1}^\top \cdots \omega_K \mathbf{1}_{n_K}^\top \right]^\top \in \mathbb{R}^n \,\middle|\, 0 \leq \omega_i \leq \omega_j \leq 1, \, \forall 1 \leq i \leq j \leq K \right\}.$$

**Step 1:**  Define a set $\bar{\mathcal{W}} := \left\{ \left[ \omega_1 \cdots \omega_n \right]^\top \in \mathbb{R}^n \,\middle|\, 0 \leq \omega_i \leq \omega_j \leq 1, \, \forall 1 \leq i \leq j \leq n \right\}$ and we have $\mathcal{W} \in \bar{\mathcal{W}}$. Given scalar gating $G_i = \begin{bmatrix} * & * \\ g_i \mathbf{1}^\top & * \end{bmatrix}$, following (11), the weighting vector returns

$$\omega := \left[ g_{1:n+1} \cdots g_{n:n+1} \right]^\top.$$

Since GLA with scalar gating valued in $[0,1]$ following Assumption 2, that is, $g_i \in [0,1]$, we have $g_{i:n+1} \leq g_{j:n+1}$ for $1 \leq i \leq j \leq n$. Therefore, any weighting vector $\omega$ implemented by GLA gating should be inside $\bar{\mathcal{W}}$.

**Step 2:**  Next, we will show that

$$\omega^\star \in \mathcal{W} \quad \text{where} \quad \omega^\star = \arg \min_{P, \omega \in \bar{\mathcal{W}}} \mathcal{L}_{\mathsf{WPGD}}(P, \omega).$$

Define the weighting vector $\omega = \left[ \omega_1^\top \cdots \omega_K^\top \right]^\top \in \mathbb{R}^n$ where we have $\omega_k = \left[ \omega_1^{(k)} \cdots \omega_{n_k}^{(k)} \right]^\top \in \mathbb{R}^{n_k}$. For any $\omega \notin \mathcal{W}$, there exist $(i, j, k)$ with $i = j - 1$ such that $\omega_i^{(k)} < \omega_j^{(k)}$. Given gradient in (41), we have that

$$\nabla_{\omega_i^{(k)}} \mathcal{L} - \nabla_{\omega_j^{(k)}} \mathcal{L} = 2 \left( M \mathrm{tr} \left( \tilde{P}^\top \tilde{P} \right) + \mathrm{tr} \left( \Sigma \tilde{P}^\top \tilde{P} \right) \right) (\omega_i^{(k)} - \omega_j^{(k)}).$$

Here, (42a) gives that (optimal) $\tilde{P} \neq 0$ and therefore, we have that $\left( M \mathrm{tr} \left( \tilde{P}^\top \tilde{P} \right) + \mathrm{tr} \left( \Sigma \tilde{P}^\top \tilde{P} \right) \right) > 0$ and $\nabla_{\omega_i^{(k)}} \mathcal{L} < \nabla_{\omega_j^{(k)}} \mathcal{L}$. Therefore either increasing $\omega_i^{(k)}$ (if $\nabla_{\omega_i^{(k)}} \mathcal{L} < 0$) or decreasing $\omega_j^{(k)}$ (if $\nabla_{\omega_j^{(k)}} \mathcal{L} > 0$) will reduce the loss. This results in showing that the optimal weighting vector $\omega^\star$ satisfies $\omega_i^{(k)} = \omega_j^{(k)}$ for any $i, j \in [n_k]$ and $k \in [K]$. Hence, $\omega^\star \in \mathcal{W}$.

**Step 3:**  Finally, we will show that any $\omega \in \mathcal{W}$ can be obtained via the GLA gating. Let $\omega = \left[ \omega_1 \mathbf{1}_{n_1}^\top \cdots \omega_K \mathbf{1}_{n_K}^\top \right]^\top$ be any vector in $\mathcal{W}$ and assume that $\omega_K = \alpha < 1$ without loss of generality. Then such sample weighting can be achieved via the gating

$$\left[ \mathbf{1}_{n_1}^\top \frac{\omega_1}{\omega_{1:K}} \cdots \mathbf{1}_{n_k}^\top \frac{\omega_k}{\omega_{k:K}} \cdots \mathbf{1}_{n_K}^\top \frac{\omega_K}{\omega_{K:K}} \right]^\top.$$

Let $\omega_k' := \frac{\omega_k}{\omega_{k:K}}$ and let $w_g$ be in the form of

$$w_g = \begin{bmatrix} \mathbf{0}_{d+1} \\ \tilde{w}_g \end{bmatrix} \in \mathbb{R}^{d+p+1}.$$

Then, it remains to show that there exists $\tilde{w}_g$ satisfying

$$\phi(\tilde{w}_g^\top \bar{c}_k) = \begin{cases} 1, & k = 0, \\ \omega_k', & k \in [K]. \end{cases}$$

The linear independence assumption in Assumption 2 implies that the problem is feasible, which completes the proof of (22).

**Proof of** (23): Recap the optimal weighting from (15) which takes the form of

$$\boldsymbol{\omega}^\star = (h_2(\gamma^\star) \cdot \boldsymbol{R} + \boldsymbol{I})^{-1} \boldsymbol{r},$$

where $h_2(\gamma^\star) > 0$. Since Assumption 3 holds and $n_1 = n_2 = \cdots = n_K := \bar{n}$, $\boldsymbol{R}$ is block diagonal matrix, where each block is an all-ones matrix. That is

$$\boldsymbol{R} = \begin{bmatrix} \mathbf{1}_{\bar{n} \times \bar{n}} & \mathbf{0} & \cdots & \mathbf{0} \\ \mathbf{0} & \mathbf{1}_{\bar{n} \times \bar{n}} & \cdots & \mathbf{0} \\ \vdots & \vdots & \ddots & \vdots \\ \mathbf{0} & \mathbf{0} & \cdots & \mathbf{1}_{\bar{n} \times \bar{n}} \end{bmatrix}.$$

Therefore, we get

$$(h_2(\gamma^\star) \cdot \boldsymbol{R} + \boldsymbol{I})^{-1} = \boldsymbol{I} - \frac{h_2(\gamma^\star)}{h_2(\gamma^\star) \cdot \bar{n} + 1} \boldsymbol{R}.$$

Since $\boldsymbol{R}\boldsymbol{r} = \bar{n}\boldsymbol{r}$, then

$$\boldsymbol{\omega}^\star = \frac{1}{h_2(\gamma^\star) \cdot \bar{n} + 1} \boldsymbol{r}.$$

Therefore, the optimal weighting (up to a scalar) is inside the set $\mathcal{W}$. Combining it with (22) completes the proof.

$\square$

### F.2 Proof of Theorem 5

*Proof.* Following the similar proof of Theorem 4, and letting $\tilde{\mathcal{W}} := \left\{ \begin{bmatrix} \omega_1 \mathbf{1}_{n_1}^\top & \cdots & \omega_K \mathbf{1}_{n_K}^\top \end{bmatrix}^\top \in \mathbb{R}^n \right\}$, we obtain

$$\min_{\boldsymbol{P} \in \mathbb{R}^{d \times d}, \boldsymbol{\omega} \in \tilde{\mathcal{W}}} \mathcal{L}_{\mathsf{WPGD}}(\boldsymbol{P}, \boldsymbol{\omega}) = \min_{\boldsymbol{P} \in \mathbb{R}^{d \times d}, \boldsymbol{\omega} \in \mathbb{R}^n} \mathcal{L}_{\mathsf{WPGD}}(\boldsymbol{P}, \boldsymbol{\omega}).$$

Therefore, it remains to show that any $\boldsymbol{\omega} \in \tilde{\mathcal{W}}$ can be implemented via some gating function. Let $\boldsymbol{\omega} = \begin{bmatrix} \omega_1 \mathbf{1}_{n_1}^\top & \cdots & \omega_K \mathbf{1}_{n_K}^\top \end{bmatrix}^\top$ be arbitrary weighting in $\tilde{\mathcal{W}}$. Theorem 4 has shown that if $\omega_1 \leq \omega_2 \leq \cdots \leq \omega_K$, GLA with scalar function can implement such increasing weighting.

Now, inspired from Appendix D, **Construction 2** and (29) that all dimensions in the output implement individual WPGD, the weighting can be a composition of up to $d + 1$ different weights ($\boldsymbol{u}$ is $(d + 1)$-dimensional). Therefore, for any $\omega_1, \cdots, \omega_K \in \mathbb{R}$, we can get $K$ separate weighting:

$$\begin{aligned} \boldsymbol{\omega}_1 &= \omega_1 [\mathbf{1}_{n_1}^\top \ \cdots \ \mathbf{1}_{n_K}^\top]^\top, \\ \boldsymbol{\omega}_2 &= (\omega_2 - \omega_1)[\mathbf{0}_{n_1}^\top \ \mathbf{1}_{n_2}^\top \ \cdots \ \mathbf{1}_{n_K}^\top]^\top, \\ \boldsymbol{\omega}_3 &= (\omega_3 - \omega_2)[\mathbf{0}_{n_1}^\top \ \mathbf{0}_{n_2}^\top \ \mathbf{1}_{n_3}^\top \ \cdots \ \mathbf{1}_{n_K}^\top]^\top, \\ &\vdots \\ \boldsymbol{\omega}_K &= (\omega_K - \omega_{K-1})[\mathbf{0}_{n_1}^\top \ \mathbf{0}_{n_2}^\top \ \mathbf{0}_{n_3}^\top \ \cdots \ \mathbf{0}_{n_{K-1}}^\top \ \mathbf{1}_{n_K}^\top]^\top. \end{aligned}$$

Recap from Appendix D and (29), and consider the construction $\tilde{\boldsymbol{W}}_v = [\mathbf{0}_{(d+p+1) \times d} \ \boldsymbol{u} \ \mathbf{0}_{(d+p+1) \times p}]^\top$. Assumption 2 implies that $K \leq p < d + p + 1$.

From (28) and (29), let $i$'th dimension implements the weighting $\omega_i$ for $i \in [K]$. Specifically, let $i$'th row of each gating matrix $\boldsymbol{G}_i$ implement weighting $[\mathbf{0}_{n_1}^\top \ \cdots \ \mathbf{0}_{n_{i-1}}^\top \ \mathbf{1}_{n_i}^\top \ \cdots \ \mathbf{1}_{n_k}^\top]$ (which is feasible due to Theorem 4) and set $u_i h_i = \omega_i - \omega_{i-1}$ with $\omega_0 = 0$. Then the composed weighting following (29) returns $\boldsymbol{\omega} = \sum_{k=1}^K \boldsymbol{\omega}_k$, which completes the proof. $\square$

### F.3 Proof of Corollary 2

*Proof.* Recap from (42a) that given $\Sigma = I$ and $\omega = 1$,

$$
P^\star = \left((d + \sigma^2)nI + (n + 1^\top R1)I\right)^{-1} 1^\top r
$$

$$
= \frac{1^\top r}{n(d + \sigma^2 + 1) + 1^\top R1} I := cI.
$$

Then taking it back to the loss function (c.f. (37)) obtains

$$
\mathcal{L}(P^\star, \omega = 1) = d + \sigma^2 - 2cd1^\top r + (d + \sigma^2)c^2 nd + (n + 1^\top R1)c^2 d
$$

$$
= d + \sigma^2 - cd1^\top r.
$$

It completes the proof. □

