# OpenReview forum: "Gating is Weighting: Understanding Gated Linear Attention through In-context Learning"
_colmweb.org/COLM/2025/Conference — COLM 2025_

### Official Review · Reviewer_Xg8p · 2025-04-30

**Rating:** 8
**Confidence:** 4
**Ethics Flag:** 1

**Summary:**

This paper contributes to the theoretical understanding of in-context learning (ICL) as optimization. Specifically, it analyzes gated linear attention showing its relationship to a form of gradient descent on the squared loss (an established construction for the study of ICL).

The theoretical results developed are nontrivial yet clearly explained, their relevance well justified, and overall they provide valuable insight into how and why in-context learning works in GLA models. The work fits in a specific niche, using established constructions and definitions, and yet to the best of my assessment distinguishes itself and goes beyond the known results in that niche.

**Questions To Authors:**

- I get that P fills the position that a preconditioning matrix would take in a gradient descent algorithm, but is P actually in some meaningful way a "preconditioner", does it reduce a condition number and/or improve convergence in some understandable sense? The form of P* in 15 suggests that maybe yes; could you please discuss/clarify?
- H3 models are referenced without defining or explaining what they are.
- In def 3, why do we call it an "effective" spectral gap, why not just a spectral gap? What does "effective" additionally denote here?
- The discussion on line 182 about multi-layer models was a bit unclear: it seems from context that previous work does not consider the casual masking scenario (does this mean they consider fully unmasked "encoder-style", or what?), and I assume it's implied that the main result here does account for casual masking, but this is not said on this line, could you clarify?
- Figure 1 is important and needs a bit of clarification. First, when skimming but before reaching the explanation for figure (d), it wasn't clear to me why GLA-vector would not be included in a-c also. I later understood (i think) that the fact that the red line (GLA) matches its dotted line is a sign that the theoretical result is reached and thus the vector formulation cannot improve anyway. However the fact that the dashed line for LinAtt and for GLA are not styled differently, this makes 1(d) more confusing. (in 1(d) it seems the GLA line matches the LinAtt theoretical bound, but it doesn't match the GLA theoretical bound, and we need the vector model for this. Is this right? It would be better if the LinAtt and GLA theoretical lines were styled differently (maybe dashed but color-coded). I recommend using opacity and line thickness in this figure to make it more readable when lines overlap a lot. The legend and the y axis can be shared to save space.
- Still on Fig 1, Is it just a coincidence or is there a reason why  in 1d GLA matches so closely LinAtt(&LinAtt optimal)?
- Still on Fig 1, is it right to infer that for a, b, and c Assumption 3 holds? Can this be checked/said explicitly? The caption could clarify why GLA-vector is only relevant in (d).
- The Vaswani citation on line 520 is wrong, missing the other authors and also inconsistent with the rest, showing only first initial instead of full first name. This suggests that all references should be double-checked and published versions cited in lieu of arxiv ones whenever the case.
- Is there some useful intuition for the form of $h_1$ and $h_2$? They look a bit like secular equations that appear in eigenvalue update problems but not quite.

**Reasons To Accept:**

- The questions and corresponding results are timely, and relevant, in the context of recent work in this direction.
- A numerical simulation study confirms the theoretical findings.
- The presentation is good, clear, and understandable to readers who are new to this line of work.

**Reasons To Reject:**

I can spot no good reasons, really. I guess this might be a more theoretical-leaning paper low on the "Empiricism, Data, and Evaluation" dimension of COLM, but this is a weak reason to reject, as the paper still is relevant to empirical work in the field.

There are a few minor points in the presentation that I list below.

---

> ### Author Response · Authors · 2025-06-03
> **Response to Reviewer Xg8p**
>
> We thank the reviewer for the positive assessment and detailed feedback.
>
> > **I can spot no good reasons, really. … relevant to empirical work in the field**.
>
> We appreciate the acknowledgment that this work fits well within COLM's scope. The reviewer correctly notes this is a "more theoretical-leaning paper," and we believe it aligns strongly with COLM's [Science of LMs](https://colmweb.org/cfp.html) track, specifically the areas of "complexity, training dynamics, grokking, learning theory for LMs." Our work makes several contributions that bridge theory and empirical understanding:
>
> **Learning Theory for LMs**: We provide the first rigorous characterization of the optimization landscape for gated linear attention models, establishing conditions under which unique global minima exist (Theorems 2-3) and connecting these architectures to interpretable learning algorithms (WPGD).
>
> **Training Dynamics Insights**: Our analysis of multi-layer GLA (Theorem 6) reveals how causal masking creates coupled gradient trajectories across layers, providing theoretical foundations for understanding training behavior in autoregressive models such as Mamba and RWKV.
>
> **Practical Impact**: By characterizing the algorithmic capabilities of different gating strategies (scalar vs. vector as provided in Theorems 4 and 5), we provide theoretical insights that directly inform architectural choices and model design decisions for practitioners working with modern efficient sequence models.
>
> > **Question to Authors**
>
> Below we address each question:
>
> - Yes, $P^\star$ in Eq.(15) has the structure of a preconditioning matrix that improves the condition number. Specifically, $P^\star= \Sigma^{-1/2}[\frac{\gamma^\star}{M} \cdot \Sigma + I]^{-1}\Sigma^{-1/2}$ effectively whitens the data covariance $\Sigma$ and applies additional regularization through the $\gamma^\star$ term. This reduces the condition number compared to the identity preconditioner, particularly when the input covariance $\Sigma$ is ill-conditioned. The factor $\gamma^\star/M$ adaptively adjusts the regularization strength based on the task correlation structure and noise level.
>
> - We will add a brief explanation: H3 ([Fu et al. 2022](https://arxiv.org/pdf/2212.14052)) is structured state space model that uses diagonal state matrices and gating mechanisms, representing an efficient alternative to transformers for long sequences.
>
> - We use "effective" to distinguish from the standard spectral gap (largest minus smallest eigenvalue) because our definition in Eq.(13) considers only nonzero eigenvalues: $\Delta_R := \lambda_{\max} - \lambda_{\min}$ where $\lambda_{\min}$ and $\lambda_{\max}$ are the **nonzero** smallest and largest eigenvalues. This is necessary since correlation matrices $R$ may have zero eigenvalues in degenerate cases.
>
> - Previous work (e.g., Ahn et al. 2024) analyzed linear attention without causal masking (i.e., full attention where each token can attend to all positions). Our multi-layer analysis (Theorem 6) accounts for causal masking where token $i$ can only attend to positions $j \leq i$, which is essential for autoregressive language modeling and introduces coupling between the gradient trajectories across layers.
>
> - Thank you for the detailed feedback on Figure 1. We will revise the figure with your suggestions: (i) Use different line styles and color-coding to distinguish LinAtt vs GLA theoretical bounds, (ii) Add opacity for overlapping lines, (iii) Share legend and y-axis, (iv) Clarify that GLA-vector is only needed in subplot (d) where Assumption 3 fails ($r_1 > r_2$), while in (a-c) scalar gating achieves the theoretical optimum.
>
> - This occurs because when Assumption 3 fails ($r_1 > r_2$), scalar gating cannot achieve the optimal WPGD solution due to the monotonicity constraint $\omega \in \mathcal{W}$. In this case, the best achievable solution for scalar gating reduces to uniform weighting $\omega = \mathbf{1}$, which is equivalent to linear attention.
>
> - Yes, Assumption 3 ($\mathbb{E}[\beta_i^\top\beta] \leq \mathbb{E}[\beta_j^\top\beta]$ for $i \leq j$) holds in subplots (a-c) where $(r_1, r_2) = (0,1), (0.2,0.8), (0.5,0.5)$ respectively, ensuring $r_1 \leq r_2$. We will add this verification to the caption.
>
> - Fixed.
>
> - These functions arise from spectral decompositions in the first-order optimality conditions and share structural similarities with secular equations. $h_2(\gamma)$ emerges from the eigendecomposition of $\Sigma$, while $h_1(\bar{\gamma})$ comes from the eigendecomposition of $R$ and captures weighted task correlations. Both have the characteristic rational form with poles at eigenvalues, similar to secular equations. Their fixed-point relationship $\gamma^* = h_1(h_2(\gamma^*)) + 1$ balances the spectral properties of data covariance and task correlation structures for optimal WPGD weighting.
>
> We again thank the reviewer for the detailed questions and suggestions, and we will incorporate them into our final manuscript.

---

> > ### Comment · Reviewer_Xg8p · 2025-06-06
> > **Thank you for the response**
> >
> > Thank you, the answers are illuminating and nothing is surprising. I maintain that this is a strong, top 50% paper and I firmly agree it is relevant to language modeling.

---

### Official Review · Reviewer_XH8t · 2025-04-30

**Rating:** 6
**Confidence:** 4
**Ethics Flag:** 1

**Summary:**

This paper investigates Gated Linear Attention (GLA) mechanisms, prevalent in modern efficient sequence models like Mamba and RWKV, through the lens of in-context learning (ICL). The core contribution is establishing a theoretical connection between multi-layer GLA and data-dependent Weighted Preconditioned Gradient Descent (WPGD) algorithms. The authors show that the gating mechanism induces data-dependent weights ω, allowing the model to modulate the influence of context tokens.

**Reasons To Accept:**

1. The paper establishes a novel and insightful connection between GLA architectures and a specific class of optimization algorithms (WPGD). This provides a valuable theoretical framework for understanding how gating contributes to the learning capabilities of models like Mamba and RWKV, going beyond standard linear attention.

2. The paper sheds light on the functional role of the gating mechanism in GLA, linking it directly to data-dependent weighting. The distinction and comparison between scalar and vector gating provide valuable insights into architectural choices and their implications for model expressivity.

**Reasons To Reject:**

1. The paper aims to understand GLA via its connection to gradient-based learning algorithms (WPGD). However, there is a significant body of existing work connecting linear attention and related recurrent models (like SSMs) to online learning. For instance, models like DeltaNet [1] frame state transitions explicitly as gradient steps minimizing an L2 loss. A more thorough discussion and comparison with these existing frameworks linking recurrence/attention to online optimization would strengthen the paper's positioning and clarify its specific contribution relative to prior art.

2. The theoretical claims are validated primarily through experiments on low-dimensional synthetic data (e.g., d=10 feature dimension) using simplified 1-to-3 layer models. While useful for illustrating the core concepts, these experiments lack scale and complexity. It remains unclear how well the specific WPGD connection and optimization landscape results translate to practical, large-scale GLA-based language models operating on high-dimensional text data. The gap between the theory/simplified experiments and real-world applications is substantial.

3. While GLA architectures are used in language models, the paper's core theoretical and experimental focus is on demonstrating ICL capabilities for linear regression tasks using synthetic data. The direct connection and implications for the complexities of *language modeling* (e.g., generative capabilities, linguistic structure understanding, large vocabulary handling) are not explicitly explored. The work feels more aligned with theoretical machine learning or representation learning venues than a conference specifically focused on language modeling. The link to language modeling is currently only asserted via the architectures studied rather than demonstrated through tasks or analysis relevant to language itself.

[1]: Yang, Songlin, Jan Kautz, and Ali Hatamizadeh. "Gated Delta Networks: Improving Mamba2 with Delta Rule." arXiv preprint arXiv:2412.06464 (2024).

---

> ### Author Response · Authors · 2025-06-03
> **Response to Reviewer XH8t - Part 1**
>
> We thank the reviewer for the thoughtful evaluation and recognition of our novel theoretical contributions. We address each concern below:
>
> > **Reason to Reject 1: The paper aims to understand GLA ... clarify its specific contribution relative to prior art.**
>
> We acknowledge the importance of positioning our work relative to existing online learning frameworks, particularly DeltaNet [1]. Recent work [[Behrouz et al. 2025a](https://arxiv.org/pdf/2505.23735)] and [[Behrouz et al. 2025b](https://arxiv.org/pdf/2504.13173)] unify sequence models as associative memory modules that optimize internal objectives through iterative algorithms, revealing connections to **online optimization and memory management dynamics**. Both DeltaNet and GLA implement variants of this online framework but with different **online learning-retaining objectives** (MSE vs dot product-based) [[Behrouz et al. 2025b](https://arxiv.org/pdf/2504.13173)]. This yields for (Gated) DeltaNet $S_t = \alpha_t (I - \beta_t k_t k_t^\top) S_{t-1} + \beta_t v_t k_t^\top$ with similarity-based forgetting (controlled by $\beta_t k_t k_t^\top$), while GLA employs $S_t = G_t \odot S_{t-1} + v_t k_t^\top$ with data-dependent gating ($G_t$). From the online learning perspective, both balance acquiring new information versus retaining past information, but GLA uses element-wise gating $G_t$ for retention while (Gated) DeltaNet combines scalar retention with content-based removal through key similarity (controlled by $\beta_t k_t k_t^\top$); please see Table 1 and Sec. 4 in [[Behrouz et al. 2025b](https://arxiv.org/pdf/2504.13173)] for further details.
>
> This online learning lens illuminates our gating↔weighting connection: the unrolling of optimization steps corresponds to context weighting. While we focus on in-context learning capabilities rather than online optimization per se, this connection helps contextualize our findings. We provide the **first rigorous optimization landscape analysis** for GLA under a heterogeneous/multitask context model, revealing how GLA attends and selects relevant context. We prove a unique global minimum (Thm. 3), classify all stationary points (Thm. 2), and specify conditions for optimality—insights absent from DeltaNet and related work. Our comparison of scalar vs. vector gating (Thms. 4–5) further uncovers expressivity trade-offs, offering principled guidance for architectural design.
>
> We will discuss the connection to [1], online learning frameworks [[Behrouz et al. 2025b](https://arxiv.org/pdf/2504.13173)], and related work in our revised manuscript.
>
> > **Reason to Reject 2: The theoretical claims are validated ... real-world applications is substantial.**
>
> Synthetic tasks have been an important proxy to explain and improve the mechanics and capabilities of more complex models. For instance [[Park et al. 2024](https://arxiv.org/pdf/2402.04248)] and [[Grazzi et al. 2024](https://arxiv.org/pdf/2402.03170)] entirely focus on comparing Mamba and Transformer on synthetic tasks.
> While we acknowledge the limitation of synthetic experiments, this conscious methodological choice aligns with established theoretical ICL literature [Ahn et al., 2024; Von Oswald et al., 2023; Mahankali et al., 2023]. Following your valuable suggestion, we **expand experiments to higher-dimensional settings**. Results follow Figure 1(b) with $(r_1,r_2)=(0.2,0.8)$ but $d=100$ instead of $d=10$:
>
> | Model | In-Context Samples ($n$) | Model Performance | Theory Prediction |
> |-------|-----------|------------------|-------------------|
> | **Linear Attention** | | | |
> | | 100 | 0.8322 | 0.8344 |
> | | 200 | 0.7525 | 0.7512 |
> | | 400 | 0.6706 | 0.6678 |
> | | 1000 | 0.5891 | 0.5840 |
> | **GLA** | | | |
> | | 100 | 0.7749 | 0.7748 |
> | | 200 | 0.6619 | 0.6617 |
> | | 400 | 0.5481 | 0.5482 |
> | | 1000 | 0.4355 | 0.4343 |
>
> **Model Performance** refers to empirical test risk achieved by training models; **Theory Prediction** refers to theoretical optimal risks: $L_{ATT}^\star$ from Corollary 3 (Eq. (25)) for Linear Attention and $L_{WPGD}^\star$ from Corollary 2 (Eq. (24)) for GLA.
>
> These results demonstrate that theoretical predictions remain highly accurate in higher-dimensional settings ($d=100$) and across varying context lengths up to $n=1000$, validating that our WPGD framework captures fundamental properties extending beyond our initial low-dimensional experiments.

---

> > ### Author Response · Authors · 2025-06-03
> > **Response to Reviewer XH8t - Part 2**
> >
> > > **Reason to Reject 3: While GLA architectures are used in language models ... analysis relevant to language itself.**
> >
> >
> > We respectfully disagree that our work lacks relevance to language modeling.
> >
> > Much of the advances in language modeling followed the seminal "Attention is all you need" paper. Arguably the key reason GLA and Mamba architectures bridge the gap between linear attention and softmax attention is their ability to implicitly attend by adaptively weighting the context window (see [[Ali et al., 2024](https://arxiv.org/pdf/2403.01590)]). GLA is a core mechanism in LLMs (e.g., Mamba, RWKV-6, GLA Transformer) that achieve performance competitive with Transformers [[Yang et al., 2024](https://arxiv.org/pdf/2312.06635); [Ali et al., 2024](https://arxiv.org/pdf/2403.01590)], and ICL is one of the most important emergent capabilities of modern LLMs.
> >
> > Our theoretical framework explains how gating enables this capability—showing how GLA architectures can **provably attend** to the context window and select relevant information through data-dependent weighting. Crucially, our theory relies on a heterogeneous context model which better captures the complexities of actual language compared to prior ICL literature that assumes single-task contexts where all tokens are equally relevant. We analyze a more realistic multitask prompting scenario [Wang et al., 2024a; Dun et al., 2023; Asai et al., 2022], where not all tokens are equally informative, and show that GLA models implicitly assign higher weight to relevant tokens. This is consistent with [[Ali et al., 2024](https://arxiv.org/pdf/2403.01590)] (see Sec. 3.1 and Fig. 5), which demonstrates that Mamba uses implicit, data-dependent gating to emphasize important features.
> > To our knowledge, we provide the first optimization-theoretic study characterizing how GLA models provably benefit from this "implicit attention" and establish separations between vector-gating, scalar-gating, and vanilla linear attention.
> >
> > As attention underlies more complex capabilities like linguistic structure understanding and tokenization, explaining implicit attention makes our contribution directly relevant to COLM and provides formal justification for observations in [[Park et al. 2024](https://arxiv.org/pdf/2402.04248)] and [[Grazzi et al. 2024](https://arxiv.org/pdf/2402.03170)] who found Mamba competitive with attention in ICL tasks.
> > Moreover, our work directly aligns with COLM's [Science of LMs](https://colmweb.org/cfp.html) track, particularly in "learning theory for LMs" and "training dynamics."

---

> > > ### Comment · Reviewer_XH8t · 2025-06-07
> > >
> > > I would like to thank the authors for their insightful rebuttal; it solves most of my existing concerns. I would like to keep my score.

---

### Official Review · Reviewer_qjca · 2025-05-20

**Rating:** 7
**Confidence:** 4
**Ethics Flag:** 1

**Summary:**

This paper provides a theoretical framework linking Gated Linear Attention (GLA) to Weighted Preconditioned Gradient Descent (WPGD). By constructing a GLA model with structured Q/K/V matrices, the authors show that GLA implements a class of WPGD, where gating acts as a weighting mechanism over in-context examples. They prove that the WPGD objective has a unique global minimum and show that GLA with scalar or vector gating can recover this optimum under certain conditions.

**Quality.** This paper introduces precise and rigorous theoretical setup and derivations. It proves connections between the optimization landscapes of GLA and WPGD. Theoretical results are complemented with empirical validations.

**Clarity.** The writing is clear and well-organized. Definitions and assumptions are clearly introduced, and contributions are outlined in relation to prior work throughout the paper.

**Originality.** The work establishes a new theoretical interpretation of GLA as implementing a general class of WPGD algorithms, expanding prior results limited to linear attention or shared-task prompts in ICL.

**Significance.** The paper addresses an important problem in understanding how gated attention architectures (e.g., Mamba, RWKV) perform ICL. By offering architectural insights, the work can shed light on how future gated models can be designed. However, the paper does assume a special attention weight structure, which might limit broader applications.

**Questions To Authors:**

1. Eqn. (9) involves specific forms of $W_k, W_q, W_v$, which may not directly reflect how real-world architectures are trained. Can the authors empirically evaluate whether a fully parameterized GLA, trained end-to-end without these structural constraints, still recovers WPGD-like behavior or achieves similar optimality?
2. While Theorem 3 establishes the uniqueness of the global minimum for the WPGD objective, can the authors discuss theoretical guarantees or insights on whether standard training procedures for GLA are guaranteed to converge to this optimum?

**Reasons To Accept:**

1. The paper addresses an important question of understanding the theoretical foundations of ICL in gated attention, which is relevant to recent architectures like Mamba and RWKV.
2. The paper rigorously establishes a novel connection between GLA and WPGD, offering a principled interpretation of how gated mechanisms enable ICL.
3. The paper adopts a multitask prompt model that allows task vectors to be correlated and not identical, extending beyond the shared-task assumption commonly made in prior ICL analyses.

**Reasons To Reject:**

1. The theoretical results rely on a structured parameterization of $W_k, W_q, W_v$ (Eq. (9)), where many entries are fixed to zero to facilitate analytical tractability. However, real-world models are trained over fully parameterized weight spaces. It is unclear whether the GLA landscape results could be extended to this more general case.
2. The paper provides the conditions for uniqueness of the global minimum (Theorem 3). It would be more informative to include convergence guarantees for training GLA models to recover this optimum.

---

> ### Author Response · Authors · 2025-06-03
> **Response to Reviewer qjca**
>
> We thank the reviewer for these questions. We address your concerns below, noting some details require verification.
> > **Q1: Eqn. (9) involves specific forms of Wk,Wq,Wv, … still recovers WPGD-like behavior or achieves similar optimality?**
>
> We clarify that our empirical experiments (see Lines 665-666) impose **no constraints on the model weights**. Indeed, despite training fully parameterized GLA models, the empirical results (solid curves in Fig. 1) align closely with our theoretical predictions (dashed curves in Fig. 1), indicating that fully parameterized GLA models implicitly exhibit WPGD-like dynamics.
>
> From a theoretical perspective, we emphasize that the restricted structure of the attention matrices is not arbitrary but is strongly supported by prior literature. For instance
>
>
> - Proposition 1 in Von Oswald et al. (2023), Section 2.4 in Lu et al. (2024), and Appendix B in Wei et al. (2024) all utilize similar structural assumptions about attention weights.
>
>
> - Theorem 1 in Ahn et al. (2024) and Proposition 1 in Li et al. (2024) demonstrate equivalence between predictions from optimized single-layer linear attention models, with and without such structural constraints.
>
>
> - Theorem 4.1 in Zhang et al. (2024) and Eq. (2) in Huang et al. (2023) show that initializing attention weights according to similar structural constraints ensures that these forms persist throughout training, with zero entries remaining zero and weights converging to forms consistent with our Eq. (9).
>
> Our study builds upon these foundations by extending the analysis to GLA models featuring nonlinear, data-dependent gating mechanisms and multitask prompting, which introduce additional complexities. Even within the structural framework provided by Eq. (9), the resulting optimization landscape is intricate and nontrivial, as thoroughly detailed in Section 4 of our paper.
>
> > **Q2: While Theorem 3 establishes the uniqueness of the global minimum for the WPGD objective,... are guaranteed to converge to this optimum?**
>
> We thank the reviewer for this insightful question. Although Theorem 3 establishes that the WPGD population risk has a uniquely attainable global minimizer with a contraction property, our analysis is limited to the optimization landscape of the WPGD problem itself. Extending these guarantees to standard training procedures for GLA introduces additional challenges, as outlined below.
>
> For the WPGD optimization directly, Theorems 2  and 3 do provide strong convergence foundations: Theorem 2 characterizes all stationary points (up to rescaling), and Theorem 3 establishes a unique global minimum (up to rescaling), together suggesting that standard optimization algorithms on WPGD population risk  $L_{WPGD}(P, \omega)$ should converge to the optimum. However, the main difficulty arises because standard  training for GLA optimizes the parameters $(W_k, W_q, W_v, \text{gating functions})$ of a GLA module, which implicitly define the WPGD parameters $(P, \omega)$ through a highly coupled and non-convex reparameterization. This **joint optimization over attention matrices and gating mechanisms** adds another layer of non-convexity absent from existing analyses of transformer optimization and the ICL–GD connection (e.g., Ahn et al., 2024; Mahankali et al., 2023). Furthermore, any rigorous convergence statement would be optimizer-specific, requiring detailed analysis of the chosen optimization method such as stochastic GD, step-size conditions, and the smoothness of the composed GLA objective.
>
> A systematic convergence analysis would ideally proceed by modeling the training dynamics over the full joint parameter space of attention matrices and gating mechanisms, and by examining how feasible regions defined by different gating schemes (cf. Theorems 4 and 5) interact with specific optimization algorithms. **Theorems 4-5 suggest a pathway where sufficiently expressive GLA parameterizations can inherit WPGD convergence properties.**  Empirically (see Fig. 1), we observe that when our theoretical conditions hold and the gating class is sufficiently expressive, standard optimizers do converge to the WPGD optimum, suggesting that the favorable landscape properties may translate to practical GLA training.
>
> However, formalizing these empirical observations would require overcoming the theoretical challenges outlined above, representing substantial future research directions beyond the current understanding of attention mechanism optimization.

---

> > ### Comment · Reviewer_qjca · 2025-06-08
> >
> > I appreciate the authors' detailed response, which addresses my main concerns. I would like to keep my original score.

---

### Official Review · Reviewer_Pxhm · 2025-05-20

**Rating:** 7
**Confidence:** 4
**Ethics Flag:** 1

**Summary:**

This paper investigates how Gated Linear Attention (GLA) models operate through the lens of In-Context Learning (ICL). It demonstrates that multi-layer GLA can execute a class of algorithms called Weighted Preconditioned Gradient Descent (WPGD), with weights that adapt dynamically based on the input data and gating mechanisms. To better illustrate how this adaptive weighting occurs, the authors introduce a multitask data model and analyzed how these WPGD-based models are optimized, proving under certain conditions the existence of a unique global minimum. Additionally, they apply these findings to single-layer GLA models, further examining their optimization landscape.
The paper also compares linear attention, scalar-gated GLA, and vector-gated GLA, highlighting the differences and explaining why gated attention models can offer superior performance compared to conventional linear attention methods.

**Questions To Authors:**

* The equivalence between GLA and WPGD (Theorem 1) and the subsequent optimization landscape analysis rely on specific constructions for attention weights (Eq. 9) and input prompts. How strong are these setups and do the authors hypothesize that similar WPGD-like dynamics could emerge implicitly in GLA models trained with practical setups?

* The problem setting, including the correlated task model (Definition 1), multitask distribution (Definition 2), and various assumptions on the problem seem too technical. Could the authors discuss how they relate to real-world in-context learning tasks?

* The paper positions GLA as a framework encompassing models like Mamba and RWKV. Could the authors elaborate further on how the specific WPGD mechanism and the data-dependent weighting in this work can be helpful to better understand these models (e.g., their particular gating functions, state update rules, or recurrence mechanisms)?

**Reasons To Accept:**

## Important and timely topic:

Efficient new LLM architectures are crucial in current research. Although GLA and similar models have shown good results in practice, their theoretical understanding is still limited. This paper contributes significantly to the therectical understanding of the gating mechanism in GLA from the ICL perspective.

## Solid theoretical framework and analysis

This paper makes a good case for connecting how GLA models work with WPGD algorithms and linking the gating mechanism and the date-dependent weighting mechasism. The interpretation of the results to ICL ablility is insightful. The optimization landscape analysis is also novel to me and adds to the theretical value of this work.

**Reasons To Reject:**

While I really appreciate that theoretical contribution of this work, my biggest concern is its practical relevance. It follows the line of work that understands ICL as doing gradient descent. However, the connection requires rather strong conditions and hand-constructed transformer weights. There are other works arguing this does not relate well to how LLMs actually do ICL [1].
Given the active research field of efficient new LLM architectures, it would be better if more practical insights/guidelines can be drawn.


[1] Shen at al. Position: Do pretrained Transformers Learn In-Context by Gradient Descent?, ICML 2024.

---

> ### Author Response · Authors · 2025-06-03
> **Response to Reviewer Pxhm - Part 1**
>
> We thank the reviewer for recognizing our theoretical contributions. We address the comments below.
>
> > **While I really appreciate that theoretical … guidelines can be drawn.**
>
> Existing works on the ICL–GD connection (Mahankali et al., 2023; Li et al., 2024b; Ahn et al., 2024) have focused on efficient architectures such as linear attention or state-space models. In contrast, real-world LLMs incorporate additional components (e.g., softmax attention, layer normalization, MLPs) that enable more sophisticated learning and lead to expected performance gaps (Shen et al., 2024). GLA or Mamba models helped bridge the gap from linear attention to transformers. Our framework enhances earlier ICL–GD settings through a richer heterogeneous context model and develops the theoretical foundations that shed light on these practically relevant architectures. Thus, while we build on the ICL–GD connection, the results and implications go further.
>
>
> Additionally, by formalizing GLA as a data-dependent weighting scheme, we show that GLA-type architectures can **provably attend and select the relevant subset of the tokens**. By characterizing **scalar vs. vector gating**, **task correlation effects on GLA expressivity**, and **optimization landscape properties**, our work also sheds light on architectural principles.  For instance, Figure 1 demonstrates clear separations between vector gating, scalar with delimiters, and scalar w/o delimiters in line with our theory.
>
> We will elaborate on these points and expand the discussion on the gap between Linear Attention-ICL (GLA-ICL in our case) and LLM-ICL raised by Shen et al. (2024) in the revision. For details on attention-weight construction and prompt design, please see our response to Q1.
>
> > **Q1:  The equivalence between GLA and WPGD… with practical setups?**
>
> Our attention weight construction and input prompt structure build on earlier theoretical studies linking ICL and preconditioned GD (Mahankali et al., 2023; Li et al., 2024b; Ahn et al., 2024). Although our theory targets specific attention weight settings, Figure 1 shows that WPGD-like dynamics still emerge in trained GLA models even when **attention weights are unconstrained** (see lines 665–666). For further details, please see our response to Reviewer **qjca**, **Q1**.
>
>
> We agree that exploring more practical prompting setup is valuable. For example, features and labels can be released sequentially in the prompt (e.g., $x_1,y_1,x_2,y_2,...$). Then, let $v_{2i}$ and $k_{2i}$ represent feature $x_i$, and $v_{2i+1}$ and $k_{2i+1}$ the label $y_i$. In this setup, the state update at time step $2i+1$ becomes:
> $$
> S_{2i+1} = G_{2i+1} \odot S_{2i} + v_{2i+1}k_{2i+1}^\top = G_{2i+1} \odot G_{2i} \odot S_{2i-1} + G_{2i} \odot v_{2i}k_{2i}^\top + v_{2i+1}k_{2i+1}^\top.
> $$
> This reduces to our Eq. (GLA) if (i) features and labels are embedded in orthogonal subspaces, and (ii) $G_{2i}$ is the all-one matrix. Finally, for more complex dataset models, a multilayer GLA model could emulate more sophisticated learners such as lasso or random forest regression (extrapolating from Garg et al. 2022, Park et al. 2024).
>
>  We will discuss these practical variations and future directions in the revision.
>
> > **Q2:  The problem setting, including the correlated … learning tasks?**
>
> Our correlated task model and multitask distribution reflect the reality that real-world prompts often include examples from multiple related tasks, not just a single one. Production systems frequently encounter heterogeneous task distributions within a single context, such as **document processing with multiple sub-tasks**, **conversational AI handling related queries**, and **educational platforms with correlated problems**. Recent works (Wang et al., 2024a; Dun et al., 2023; Asai et al., 2022; Lines 279–281) have documented these scenarios and proposed methods for multitask prompts. Our framework provides the first theoretical analysis of how GLA's data-dependent weighting addresses such scenarios. We will expand on this discussion in the revision.

---

> > ### Author Response · Authors · 2025-06-03
> > **Response to Reviewer Pxhm - Part 2**
> >
> > > **Q3: The paper positions GLA as a framework …  or recurrence mechanisms)?**
> >
> > Our WPGD framework provides concrete insights into the mechanisms behind GLA-based models by revealing the optimization principles underlying their design.
> > Recent theoretical work offers complementary views. For example, [[Ali et al., 2024](https://arxiv.org/pdf/2403.01590)] show that Mamba embeds implicit attention matrices via data-controlled linear operators, while [[Sieber et al., 2024](https://arxiv.org/pdf/2405.15731)] unify SSMs, attention, and RNNs within a dynamical-systems framework. Additionally, Table 1 in [[Yang et al., 2023](https://arxiv.org/pdf/2312.06635)] outlines the gating functions used by various models such as Mamba and RWKV.
> >
> > In line with these studies, our WPGD–ICL framework provides the optimization foundation for understanding when data-dependent weighting yields optimal WPGD learning and guarantees a unique global minimum.
> > For Mamba, our framework clarifies how selective state-space mechanisms enable data-dependent weighting that adapts to context, grounding GLA and Mamba’s empirical advantages over non-selective SSMs in theory. For RWKV, which uses gating and channel-mixing, our framework explains how these choices affect achievable weightings. Our analysis of scalar vs. vector gating (Theorems 4–5) guides when each design achieves optimal WPGD, illuminating architectural trade-offs.
> >
> > We will expand on these points in the revision.

---

> > > ### Comment · Reviewer_Pxhm · 2025-06-08
> > >
> > > I thank the authors for their detailed response. Most of my concerns have been addressed and I would like to keep my rating with increased confidence.

---

### Decision · Program_Chairs · 2025-07-08

**Decision:**

Accept

**Comment:**

This paper explores the gated linear attention mechanism (popularized by models like Mamba) through a largely theoretical lens. The reviewers are unanimously positive about the paper, and were engaged in a lively discussion with the authors. it appears to be a solid contribution on an important topic. The primary concerns raised were about the strong (and perhaps unrealistic) assumptions that were made in the theoretical components in order to provide provable guarantees. However, for theoretical work, this is often the case, and it is nonetheless valuable to start somewhere. Overall, this work would be nice to see at COLM.